# On the Computational Landscape of Replicable Learning

**Alkis Kalavasis**
**Yale University**
alvertos.kalavasis@yale.edu

**Amin Karbasi**
**Yale University**
amin.karbasi@yale.edu

**Grigoris Velegkas**
**Yale University**
grigoris.velegkas@yale.edu

**Felix Zhou**
**Yale University**
felix.zhou@yale.edu

## Abstract

We study computational aspects of algorithmic replicability, a notion of stability introduced by Impagliazzo, Lei, Pitassi, and Sorrell [2022]. Motivated by a recent line of work that established strong *statistical* connections between replicability and other notions of learnability such as online learning, private learning, and SQ learning, we aim to understand better the *computational* connections between replicability and these learning paradigms. Our first result shows that there is a concept class that is efficiently replicably PAC learnable, but, under standard cryptographic assumptions, no efficient online learner exists for this class. Subsequently, we design an efficient replicable learner for PAC learning parities when the marginal distribution is far from uniform, making progress on a question posed by Impagliazzo et al. [2022]. To obtain this result, we design a replicable lifting framework inspired by Blanc, Lange, Malik, and Tan [2023] that transforms in a black-box manner efficient replicable PAC learners under the uniform marginal distribution over the Boolean hypercube to replicable PAC learners under any marginal distribution, with sample and time complexity that depends on a certain measure of the complexity of the distribution. Finally, we show that any pure DP learner can be transformed to a replicable one in time polynomial in the accuracy, confidence parameters and exponential in the representation dimension of the underlying hypothesis class.

## 1 Introduction

The replicability crisis is omnipresent in many scientific disciplines including biology, chemistry, and, importantly, AI [Baker, 2016, Pineau et al., 2019]. A recent article that appeared in Nature [Ball, 2023] explains how the reproducibility crisis witnessed in AI has a cascading effect across many other scientific areas due to its widespread applications in other fields, like medicine. Thus, a pressing task is to design a formal framework through which we can argue about the replicability of experiments in ML. Such an attempt was initiated recently by the pioneering work of Impagliazzo et al. [2022], who proposed a definition of replicability as a property of learning algorithms.

**Definition 1.1** (Replicable Algorithm; Impagliazzo et al., 2022). *Let $\mathcal{R}$ be a distribution over random strings. A learning algorithm $\mathcal{A}$ is $n$-sample $\rho$-replicable under distribution $\mathcal{D}$ if for two independent sets $S, S' \sim \mathcal{D}^n$ it holds that $\Pr_{S,S'\sim\mathcal{D}^n, r\sim\mathcal{R}}[\mathcal{A}(S, r) \neq \mathcal{A}(S', r)] \leq \rho$. We will say that $\mathcal{A}$ is replicable if the above holds uniformly over all distributions $\mathcal{D}$.*

We emphasize that the random string $r$ is shared across the two executions and this aspect of the definition is crucial in designing algorithms that have the same output under two different i.i.d. inputs. Indeed, both Dixon et al. [2023] and Chase et al. [2023b] demonstrated learning tasks for which there

38th Conference on Neural Information Processing Systems (NeurIPS 2024).

are no algorithms that satisfy this strong notion of replicability when the randomness is not shared across executions. This shared random string models the random seed of a learning algorithm in practice, and sharing internal randomness can be easily implemented by sharing said random seed.

Closer to our work, Impagliazzo et al. [2022], Ghazi et al. [2021b], Bun et al. [2023], Kalavasis et al. [2023] established strong *statistical* connections between replicability and other notions of algorithmic stability and learning paradigms such as differential privacy (DP), statistical queries (SQ), and online learning. Although several of these works provided various *computational* results, these connections are not as well understood as the statistical ones.

## 1.1 Our Contributions

In this work, we aim to shed further light on the aforementioned *computational* connections. In particular, we provide both negative and positive results. Negative results are manifested through *computational separations* while positive results are presented through *computational transformations*, as we will see shortly. We emphasize that whenever we work within the PAC learning framework we consider the *realizable* setting.

**Replicability & Online Learning.** The results of Ghazi et al. [2021b], Bun et al. [2023], Kalavasis et al. [2023] established a statistical connection between replicability and online learning. In particular, in the context of PAC learning, these works essentially show that replicable PAC learning is statistically equivalent to online learning since learnability in both settings is characterized by the finiteness of the Littlestone dimension of the underlying concept class. From the above, the first natural question is the following:

Q1. *How does replicability computationally relate to online learning?*

We show that under standard cryptographic assumptions, efficient replicability is separated from efficient online learning.

**Theorem 1.2** (Informal, see Theorem 2.1). *Assuming the existence of one-way functions, there is a concept class that is replicably PAC learnable in polynomial time, but is not efficiently online learnable by any no-regret algorithm.*

In order to prove the above result, we provide an efficient replicable PAC learner for the concept class of *One-Way Sequences* $\mathcal{OWS}$ over $\{0,1\}^d$, introduced by Blum [1994]. This algorithm, which relies on a novel replicable subroutine for quantile estimation (cf. Theorem B.6), combined with the cryptographic hardness result of Bun [2020] for online learnability, gives the desired computational separation. For further details, we refer to Section 2.

**Replicability & SQ.** The Statistical Query (SQ) framework is an expressive model of statistical learning introduced by Kearns [1998]. This model of learning is a restricted class of algorithms that is only permitted indirect access to samples through approximations of certain carefully chosen functions of the data (statistical queries). The motivation behind the introduction of the SQ framework was to capture a large class of noise-resistant learning algorithms. As such, any SQ algorithm enjoys some inherent stability properties. Since replicability also captures some notion of algorithmic stability, it is meaningful to ask:

Q2. *How does replicability computationally relate to SQ learning?*

For some background about the SQ framework, we refer the reader to Appendix C.2. A manifestation of the intuitive connection between SQ and replicability can be found in one of the main results of Impagliazzo et al. [2022], which states that any efficient SQ algorithm can be efficiently made replicable. In this paper, we investigate the other direction of this connection.

A particularly interesting computational separation between (distribution-specific) replicability and SQ learning was noticed by Impagliazzo et al. [2022], in the context of realizable binary classification: if we consider the uniform distribution $\mathcal{U}$ over the Boolean hypercube $\{0,1\}^d$, then the concept class of parities is SQ-hard to learn under $\mathcal{U}$ [Kearns, 1998] but admits an efficient PAC learner that is replicable under $\mathcal{U}$. Indeed, with high probability over the random draw of the data, Gaussian elimination, which is the standard algorithm for PAC learning parities under $\mathcal{U}$, gives a *unique* solution and, as a result, is replicable.

Based on this observation, Impagliazzo et al. [2022] posed as an interesting question whether parities are efficiently learnable by a replicable learner under other marginal distributions, for which Gaussian elimination is "unstable", i.e., it fails to give a unique solution (see e.g., Proposition C.4).

**Transforming Distribution-Specific Replicable Learners.** Inspired by the question of Impagliazzo et al. [2022], we ask the following more general question for realizable binary classification:

Q3. *For a class $\mathscr{C} \subseteq \{0,1\}^{\mathcal{X}}$ and marginal distributions $\mathcal{D}_1, \mathcal{D}_2$ over $\mathcal{X}$, how does replicable PAC learnability of $\mathscr{C}$ under $\mathcal{D}_1$ and under $\mathcal{D}_2$ relate to one another computationally?*

Our main result is a general black-box transformation from a replicable PAC learner under the uniform distribution over $\mathcal{X} = \{0,1\}^d$ to a replicable PAC learner under some unknown marginal distribution $\mathcal{D}$. The runtime of the transformation depends on the *decision tree complexity* of the distribution $\mathcal{D}$, a complexity measure that comes from the recent work of Blanc et al. [2023] and is defined as follows:

**Definition 1.3** (Decision Tree Complexity; Blanc et al., 2023)**.** *The decision tree complexity of a distribution $\mathcal{D}$ over $\{0,1\}^d$ is the smallest integer $\ell$ such that its probability mass function (pmf) can be computed by a depth-$\ell$ decision tree (cf. Definition D.1).*

A *decision tree* (cf. Definition D.1) $T : \{0,1\}^d \to \mathbb{R}$ is a binary tree whose internal nodes query a particular coordinate $x_i, i \in [d]$ (descending left if $x_i = 0$ and right otherwise) and whose leaves are labelled by real values. Each $x \in \{0,1\}^d$ follows a unique root-to-leaf path based on the queried coordinate of internal nodes, and its value $T(x)$ is the value stored at the root. For some $\mathcal{D}$ over the Boolean hypercube, we say $\mathcal{D}$ is *computed by a decision tree $T$ of depth $\ell$* if, for any $x \in \{0,1\}^d$, it holds that $\mathcal{D}(x) = T(x)$.

As an example, note that the uniform distribution requires depth $\ell = 1$ since its pmf is constant while a distribution whose pmf takes $2^d$ different values requires depth $\ell = d$. However, various natural (structured) distributions have tree complexity much smaller than $d$. Importantly, the running time overhead of our lifting approach scales proportionally to the decision tree complexity of $\mathcal{D}$ and hence can be used to obtain novel efficient replicable PAC learners.

Our replicable lifting framework informally states the following:

**Theorem 1.4** (Informal, see Theorem 3.2)**.** *Let $\mathcal{X} = \{0,1\}^d$ and $m = \mathrm{poly}(d)$. Consider a concept class $\mathscr{C} \subseteq \{0,1\}^{\mathcal{X}}$ and assume that $\mathcal{A}$ is a PAC learner for $\mathscr{C}$ that is $m$-sample replicable under the uniform distribution and runs in time $\mathrm{poly}(m)$. Then, for some (unknown) monotone[1] distribution $\mathcal{D}$ with decision tree complexity $\ell$, there exists an algorithm $\mathcal{A}'$ that is a PAC learner for $\mathscr{C}$ that is $(m\ell)^{\ell}$-sample replicable under $\mathcal{D}$ and runs in time $\mathrm{poly}((m\ell)^{\ell})$.*

This implies for instance that for $\ell = O(1)$, the blowup is polynomial in the runtime of the uniform learner while if $\ell = O(\log d)$, the running time is quasi-polynomial in $d$. We note that our result can also be extended to general (non-monotone) distributions $\mathcal{D}$ if we assume access to a *(subcube) conditional sampling oracle*. For formal details, we refer the reader to Section 3.

As an application of this framework, we show how to black-box transform replicable learners for parities that work under the uniform distribution to replicable learners that work under some other unknown distribution $\mathcal{D}$, where the sample complexity and running time of the transformation depends on the decision tree complexity of $\mathcal{D}$. This result is hence related to the question of Impagliazzo et al. [2022] since we can design distributions $\mathcal{D}$ over $\{0,1\}^d$ with small decision tree complexity for which Gaussian elimination fails to be replicable. We refer to Section 4 for the formal statement.

**Corollary 1.5** (Informal, see Corollary 4.3 and Theorem C.5)**.** *Let $\mathcal{X} = \{0,1\}^d$ and $m = \mathrm{poly}(d)$. Then, for any (unknown) monotone distribution $\mathcal{D}$ with decision tree complexity $\ell$, there exists a PAC learner for parities that is $(m\ell)^{\ell}$-sample replicable under $\mathcal{D}$ and runs in time $\mathrm{poly}((m\ell)^{\ell})$.*

*Moreover, for any $m' = m^{\Theta(1)}$, there exists some monotone distribution $\mathcal{D}$ over $\{0,1\}^d$ with decision tree complexity $\ell = \Theta(1)$ so that Gaussian elimination, with input $m'$ labeled examples, fails to PAC learn parities replicably under $\mathcal{D}$ with constant probability.*

---

[1] Recall that $\mathcal{D}$ over $\{0,1\}^d$ is monotone if whenever $x \succeq y$ in the partial order of the poset, we have $\mathcal{D}(x) \geq \mathcal{D}(y)$.

**Replicability & Privacy.** Returning to the works of Ghazi et al. [2021b], Bun et al. [2023], Kalavasis et al. [2023], it is known that replicable PAC learning is statistically equivalent to approximate differentially private PAC learning. We hence ask:

Q4. *How does replicability computationally relate to private learning?*

The recent work of Bun et al. [2023] provided a computational separation between *approximate* differential privacy and replicability. In particular, they showed that a replicable learner can be efficiently transformed into an approximately differentially private one, but the other direction is hard under standard cryptographic assumptions. To be more specific, they provided a concept class that is efficiently learnable by an approximate DP learner, but not efficiently learnable by a replicable learner, assuming the existence of one-way functions. Based on this hardness result, we ask whether we can transform a *pure* DP algorithm into a replicable one. We show the following result.

**Theorem 1.6** (Informal; see Theorem 5.2). *Let $\mathcal{D}_{XY}$ be a distribution on $\mathcal{X} \times \{0,1\}$ that is realizable with respect to some concept class $\mathscr{C}$. Let $\mathcal{A}$ be an efficient pure DP learner for $\mathscr{C}$. Then, there is a replicable learner $\mathcal{A}'$ for $\mathscr{C}$ that runs in time polynomial with respect to the error, confidence, and replicability parameters but exponential in the representation dimension[2] of $\mathscr{C}$.*

We reiterate that this transformation is efficient with respect to the correctness, confidence and replicability parameters $\alpha, \beta, \rho$, but *it could be inefficient* with respect to some parameter that captures the complexity of the underlying concept class $\mathscr{C}$ (such as the representation dimension of the class, e.g., the margin parameter in the case of large-margin halfspaces). To the best of our knowledge, the same holds for the transformation of a pure DP learner to an online learner of Gonen et al. [2019] and it would be interesting to show that this is unavoidable.

**The Computational Landscape of Stability.** Our work studies several computational aspects of replicability and its connections with other stability notions such as online learning, SQ learning, and differential privacy. Combining our results with the prior works of Blum et al. [2005], Gonen et al. [2019], Ghazi et al. [2021b], Impagliazzo et al. [2022], Bun et al. [2023], Kalavasis et al. [2023] yields the current computational landscape of stability depicted below.

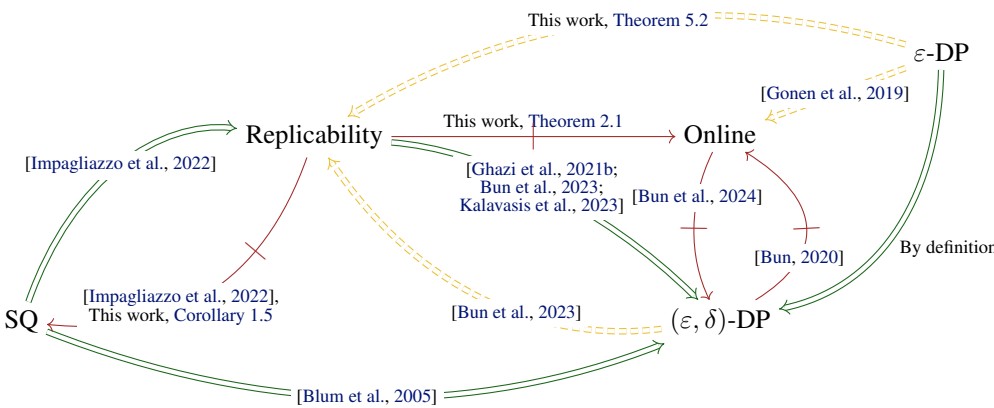

---

[2]The *representation dimension* is a combinatorial dimension, similar to VC dimension, that characterizes which classes are PAC learnable by pure DP algorithms [Beimel et al., 2013].

Figure 1.1: The computational landscape of stability. A green double arrow ($\Rightarrow$) from tail to head indicates that an efficient learner for a task in the setting of the arrow tail can be black-box transformed into an efficient learner for the same task in the setting of the arrow head. Meanwhile, an orange dashed double arrow ($==\succ$) from tail to head indicates that an efficient learner for a task in the tail setting can be black-box transformed into an efficient learner for the same task in the head setting, under some additional assumptions. Finally, a red slashed single arrow ($\nrightarrow$) from tail to head indicates that there is a learning task for which an efficient learner exists in the setting of the arrow tail but no efficient learner can exist in the setting of the arrow head, possibly under some cryptographic assumptions.

The computational landscape of "stable" learning depicted in Figure 1.1 is a byproduct of the following results connecting (i) replicability, (ii) approximate DP, (iii) pure DP, (iv) online learning, and, (v) statistical queries.

**Black-Box Transformations.**
1. Pure DP can be efficiently[3] transformed to Online [Gonen et al., 2019].
2. Replicability can be efficiently transformed to Approximate DP [Ghazi et al., 2021b, Bun et al., 2023, Kalavasis et al., 2023].
3. SQ can be efficiently transformed to Approximate DP [Blum et al., 2005].[4]
4. SQ can be efficiently transformed to Replicable [Impagliazzo et al., 2022].
5. Pure DP can be efficiently[5] transformed to Replicable (this work, Theorem 5.2).

**Caveats of Transformations.** The transformations from pure $\varepsilon$-DP learners to replicable and online learners may incur exponential computation time in the *representation dimension* [Beimel et al., 2013] of the underlying hypothesis class.

Regarding the approximate DP reduction to replicability: The transformation provided by Bun et al. [2023] uses correlation, which necessitates that the output space of the algorithm is finite. To be more precise, based on [Bun et al., 2023], there is an efficient transformation from approximate DP to perfectly generalizing algorithms. Next, the authors use correlated sampling to obtain a replicable learner as follows. Given a perfectly generalizing algorithm $A$ and sample $S$, the correlated sampling strategy is applied to the distribution of outputs of $A(S)$. Hence, the output space of $A$ should be finite. In the PAC learning setting, to ensure that the algorithm has finite output space, one sufficient condition is that the domain $\mathcal{X}$ is finite. The correlated sampling step can be explicitly implemented via rejection sampling from the output space of $A$. The acceptance probability is controlled by the probability mass function of $A(S)$. As a result, in general, it is not computationally efficient. For instance, if the finite input space to the correlated sampling strategy is $\{0,1\}^d$, then the runtime of the algorithm could be $\exp(d)$, since the acceptance probability is exponentially small in the dimension in the worst case.

For the specific case of PAC learning, there is another transformation from approximate DP to replicability that holds for countable domains $\mathcal{X}$ that was proposed by Kalavasis et al. [2023], but this approach goes through the Littlestone dimension of the class and might not even be computable in its general form.

**Separations.** There is a concept class that can be learned efficiently
1. by a replicable PAC learner (this work, Theorem 2.1), but not an efficient online one under OWF [Blum, 1994];
2. by an approximate DP PAC learner, but not an efficient online one under OWF [Bun, 2020];
3. by an online learner, but not an efficient approximate DP PAC one under cryptographic assumptions[6] [Bun et al., 2024];
4. by a replicable PAC learner under uniform marginals (Impagliazzo et al. [2022]) or more general marginals (this work, Corollary 1.5), but not an efficient SQ one;

---

[3]The transformation is efficient with respect to the correctness and confidence parameters but could be inefficient with respect to other parameters of the class such as the representation dimension.

[4]The reduction is efficient as it simply requires adding Gaussian noise to each statistical query. However, it is impractical for most tasks of interest as the resulting privacy guarantees are relatively weak.

[5]The transformation is efficient with respect to the correctness and confidence parameters but could be inefficient with respect to other parameters of the class.

[6]The specific assumptions are technical and we omit the specific statements. Roughly, the assumptions lead to the possibility to build indistinguishability obfuscation for all circuits.

5. by an approximate DP PAC learner, but not an efficient replicable one under OWF [Bun et al., 2023].

## 1.2 Related Work

**Replicability.** Pioneered by Impagliazzo et al. [2022], there has been a growing interest from the learning theory community in studying replicability as an algorithmic property. Esfandiari et al. [2023a,b] studied replicable algorithms in the context of multi-armed bandits and clustering. Later, Eaton et al. [2023], Karbasi et al. [2023] studied replicability in the context of Reinforcement Learning (RL) and designed algorithms that achieve various notions of replicability. Recently, Bun et al. [2023] established statistical equivalences and separations between replicability and other notions of algorithmic stability such as differential privacy when the domain of the learning problem is finite and provided some computational and statistical hardness results to obtain these equivalences, under cryptographic assumptions. Subsequently, Kalavasis et al. [2023] proposed a relaxation of the replicability definition of Impagliazzo et al. [2022], showed its statistical equivalence to the notion of replicability for countable domains[7] and extended some of the equivalences from Bun et al. [2023] to countable domains. Chase et al. [2023b], Dixon et al. [2023] proposed a notion of *list-replicability*, where the randomness is not shared across the executions of the algorithm, but the outputs are required to belong to a list of small cardinality instead of being identical. Both of these works developed algorithms that work in the *realizable* setting, and later Chase et al. [2023a] showed that, surprisingly, it is impossible to design list-replicable learning algorithms for infinite classes in the *agnostic* setting. Recently, Moran et al. [2023] established even more statistical connections between replicability and other notions of stability, by dividing them into two categories consisting of distribution-dependent and distribution-independent definitions. Recent work by Kalavasis et al. [2024] studies replicable algorithms for large-margin halfspaces.

**Computational Separations & Transformations in Stability.** The seminal result of Blum [1994] illustrated a computational separation between PAC and online learning. Later, Bun [2020] obtained a similar separation between private PAC and online learning. More recently, Bun et al. [2023] showed that there exists a concept class that admits an efficient approximate DP learner but not an efficient replicable one, assuming the existence of OWFs. Contributing to this line of work, our Theorem 2.1 is of similar flavor.
Shifting our attention to SQ learning, there is an intuitive similarity between learning from noisy examples and private learning: algorithms for both problems must be robust to small variations in the data (see also Blum et al. [2005]). Parities are a canonical SQ-hard problem, meaning that no efficient SQ algorithm for this class exists. Kasiviswanathan et al. [2011] designed an efficient private learner for learning parities, which dispels the similarity between learning with noise and private learning. Georgiev and Hopkins [2022] showed that while polynomial-time private algorithms for learning parities are known, the failure probability of any such algorithm must be larger than what can be achieved in exponential time, or else $\mathrm{NP} = \mathrm{RP}$. Finally, recent work by Bun et al. [2024] gives a concept class that admits an online learner running in polynomial time with a polynomial mistake bound, but for which there is no computationally efficient approximate differentially private PAC learner, under cryptographic assumptions.
Moving on to transformations, our Theorem 1.4 builds upon the result of Blanc et al. [2023] who designed a framework that transforms computationally efficient algorithms under uniform marginal distributions, to algorithms that work under some other distribution, where the complexity of the transformation scales with some particular notion of distance between the two distributions. Essentially, our result can be viewed as a replicable framework of the same flavor, with a small additional computational and statistical overhead that scales with the replicability parameter. Finally, our approach to transform a pure DP learner into a replicable learner (cf. Theorem 1.6) is inspired by Gonen et al. [2019], who provided a transformation from pure DP learners to online learners.

## 1.3 Notation

In general, we use $\mathcal{A}$ to denote an algorithm. For unsupervised problems, we usually denote by $\mathcal{D}$ the distribution over input examples. In the case of supervised problems, we use $\mathscr{C}$ to denote the concept class in question, $\mathcal{D}$ to denote the marginal distribution of the feature domain $\mathcal{X}$, and $\mathcal{D}_{\mathcal{X}\mathcal{Y}}$ to refer to the joint distribution over labeled examples $(x, y) \in \mathcal{X} \times \mathcal{Y}$. Throughout our work, we use $\alpha, \beta$ to refer to the error and failure probability parameters of the algorithm, $\varepsilon, \delta$ to denote the approximate

---

[7]We remark that this equivalence for finite domains can also be obtained, implicitly, from the results of Bun et al. [2023]. Connections between replicability and the Littlestone dimension go back to Ghazi et al. [2021b].

DP parameters, and $\rho$ for the replicability parameter. In most cases, the feature domain $\mathcal{X}$ is a subset of a high-dimensional space and we use $d$ to denote the dimension of that space.

## 2 Efficient Replicability and Online Learning

Our first main result shows that replicability and online learning are *not* computationally equivalent, assuming the existence of one-way functions. This hardness result is based on a construction from Blum [1994], who defined a concept class denoted by $\mathcal{OWS}$, which is efficiently PAC learnable but not online learnable in polynomial time, assuming the existence of one-way functions. Blum's construction builds upon the Goldreich-Goldwasser-Micali pseudorandom function generator [Goldreich et al., 1986] to define families of "one-way" labeled sequences $(\sigma_1, b_1), \ldots, (\sigma_r, b_r) \in \{0,1\}^d \times \{0,1\}$, for some $r = \omega(\text{poly}(d))$. These string-label pairs can be efficiently computed in the forward direction, but are hard to compute in the reverse direction. Specifically, for any $i < j$, it is easy to compute $\sigma_j, b_j$ given $\sigma_i$. On the other hand, it is hard to compute $\sigma_i, b_i$ given $\sigma_j$.

The essence of the difficulty in the online setting is that an adversary can present to the learner the sequence in reverse order, i.e., $\sigma_r, \sigma_{r-1}, \ldots, \sigma_1$. Then, the labels $b_i$ are not predictable by a polynomial time learner. However, in the PAC setting, for any distribution over the sequence $\{(\sigma_i, b_i)\}_{i \in [r]}$, a learner which is given $n$ labeled examples can identify the string $\sigma_{i^*}$ with smallest index $i^*$ in the sample. Then, it can perfectly and efficiently predict the label of any string that comes after it. This approximately amounts to a $(1 - 1/n)$ fraction of the underlying population.

It is not hard to see that the PAC learner for $\mathcal{OWS}$ is not replicable, since the minimum index of a sample can vary wildly between samples. The high-level idea of our approach to making the algorithm replicable is to show that we can replicably identify an *approximate* minimum index and output the hypothesis that (efficiently) forward computes from that index. Our main result, proven in Appendix B, is as follows.

**Theorem 2.1.** *Let $d \in \mathbb{N}$. The following hold:*

*(a) [Bun, 2020] Assuming the existence of one-way functions, the concept class $\mathcal{OWS}$ with input domain $\{0,1\}^d$ cannot be learned by an efficient no-regret algorithm, i.e., an algorithm that for some $\eta > 0$ achieves expected regret $\mathbb{E}[R_T] \leq \text{poly}(d) \cdot T^{1-\eta}$ using time $\text{poly}(d, T)$ in every iteration.[8]*

*(b) (Theorem B.7) The concept class $\mathcal{OWS}$ with input domain $\{0,1\}^d$ can be learned by a $\rho$-replicable $(\alpha, \beta)$-PAC learner with sample complexity $m = \text{poly}(d, 1/\alpha, 1/\rho, \log(1/\beta))$ and $\text{poly}(m)$ running time.*

## 3 Lifting Replicable Uniform Learners

In this section, we present our replicable lifting framework in further detail. First, we will need the following technical definition.

**Definition 3.1** (Closed under Restrictions). *A concept class $\mathscr{C}$ of functions $f : \{0,1\}^d \to \{0,1\}$ is closed under restrictions if, for any $f \in \mathscr{C}, i \in [d]$ and $b \in \{0,1\}$, the restriction $f_{i=b}$ remains in $\mathscr{C}$, where $f_{i=b}(x) = f(x_1, ..., x_{i-1}, b, x_{i+1}, ..., x_d)$ for any $x \in \{0,1\}^d$.*

Our replicable lifting result works for concept classes that satisfy the closedness under restrictions property. Also, recall that a distribution $\mathcal{D}$ over $\{0,1\}^d$ is *monotone* (Definition D.4) if whenever $x \succeq y$ ($x$ is greater than $y$ in the partial ordering of the poset), it holds $\mathcal{D}(x) \geq \mathcal{D}(y)$. Our algorithm for lifting replicable uniform learners to replicable learners under some unknown distribution $\mathcal{D}$ is presented in Theorem 3.2. This lifting approach is inspired by the work of Blanc et al. [2023], who designed the non-replicable variant of this transformation. We note that our algorithm, similar to the algorithm of Blanc et al. [2023], only requires sample access to $\mathcal{D}$ for monotone distributions, while, for arbitrary non-monotone probability measures, our algorithm requires access to a *conditional sampling oracle* (cf. Definition D.5). Our replicable lifting theorem reads as follows.

**Theorem 3.2** (Lifting Replicable Uniform Learners). *Consider a concept class $\mathscr{C}$ of functions $f : \{0,1\}^d \to \{0,1\}$ closed under restrictions. Suppose we are given black-box access to an algorithm such that for any $\alpha', \rho', \beta' \in (0,1)$, given $\text{poly}(d, 1/\alpha', 1/\rho', \log(1/\beta'))$ samples from $\mathcal{U}$,*

  *(i) is $\rho'$-replicable with respect to the uniform distribution $\mathcal{U}$,*

  *(ii) PAC learns $\mathscr{C}$ under the uniform distribution to accuracy $\alpha'$ and confidence $\beta'$, and,*

  *(iii) terminates in time $\text{poly}(d, 1/\alpha', 1/\rho', \log(1/\beta'))$.*

---

[8]We remark that this stronger requirement in regret is necessary since the trivial random guessing algorithm achieves $o(T)$ regret in finite domains (cf. Appendix A.3).

*Let $\alpha, \rho \in (0,1)$ and $\beta \in (0, \rho/3)$. For $m = \mathrm{poly}(d, 1/\alpha, 1/\rho, \log(1/\beta))$ and $M = \mathrm{poly}(d, 1/\alpha, 1/\rho, \log(1/\beta))^{O(\ell)}$, the following cases hold:*

(a) *If $\mathcal{D}$ is a monotone distribution over $\{0,1\}^d$ representable by a depth-$\ell$ decision tree, there is an algorithm that draws $M$ samples from $\mathcal{D}$, is $\rho$-replicable with respect to $\mathcal{D}$, PAC learns $\mathscr{C}$ under $\mathcal{D}$ with accuracy $\alpha$ and confidence $\beta$, and terminates in $\mathrm{poly}(M)$ time.*

(b) *If $\mathcal{D}$ is an arbitrary distribution over $\{0,1\}^d$ representable by a depth-$\ell$ decision tree, there is an algorithm that draws $M$ labeled examples as well as $M$ conditional samples (cf. Definition D.5) from $\mathcal{D}$, is $\rho$-replicable with respect to $\mathcal{D}$, PAC learns $\mathscr{C}$ with respect to $\mathcal{D}$ with accuracy $\alpha$ and confidence $\beta$, and terminates in $\mathrm{poly}(M)$ time.*

The high-level idea of the reduction proceeds as follows. Let us consider the realizable PAC setting, i.e., the labels are consistent with some $f^\star \in \mathscr{C}$. Let us assume black-box access to a replicable uniform learner $\mathcal{A}_{\mathscr{C}}^{\mathcal{U}}$ for $\mathscr{C}$ and access to i.i.d. samples of the form $(x, f^\star(x))$, where $x \sim \mathcal{D}$. Our goal is to (efficiently) obtain an algorithm $\mathcal{A}_{\mathscr{C}}^{\mathcal{D}}$ that achieves small misclassification error with respect to the unknown distribution $\mathcal{D}$ and target $f^\star$ and is also replicable under $\mathcal{D}$. The promise is that $\mathcal{D}$ has a decision tree representation of depth $\ell$.[9]

The first step is to draw enough samples $(x, f^\star(x)), x \sim \mathcal{D}$, and compute the decision tree representation of $\mathcal{D}$. This is an unsupervised learning task and it only uses the feature vectors of the training set. We design a replicable algorithm for this step (cf. Theorem E.7), which could be of independent interest. The next observation is crucial: any discrete distribution on $\{0,1\}^d$ can be expressed as a mixture of uniform distributions conditioned on non-overlapping sub-cubes. To see this, notice that any root-to-leaf path in the estimated decision tree representation corresponds to a sub-cube of $\{0,1\}^d$ and, conditioned on this path, the remaining coordinates follow a uniform law. Hence, after an appropriate re-sampling procedure, one can employ the black-box replicable learner $\mathcal{A}_{\mathscr{C}}^{\mathcal{U}}$ to any one of the leaves $t$ of the tree decomposition of $\mathcal{D}$ and obtain a classifier $f_t$. For this step, the fact that $\mathscr{C}$ is closed under restrictions is crucial. Intuitively, if we wish to implement this idea in a replicable manner and need overall replicability parameter $\rho$, it suffices to use the uniform replicable algorithm in each leaf with parameter $O(\rho/2^\ell)$ since we make at most $2^\ell$ calls to the uniform PAC learner (the decision tree complexity of the target is $\ell$). Finally, given a test example $x \sim \mathcal{D}$, one computes the leaf $t$ that corresponds to the sub-cube that $x$ falls into and uses the (replicable) output $f_t$ of the associated uniform PAC learner to guess the correct label.

For the formal analysis, we refer to Appendix D and Appendix E.

## 4 Efficient Replicability and SQ Learning: Parities

In this section, we provide an application of the general lifting framework we described in Section 3. One of the main results of the seminal work of Impagliazzo et al. [2022] is that any SQ algorithm can be made replicable. This result allows a great collection of tasks to be solved replicably since the SQ framework is known to be highly expressive [Kearns, 1998].

The primary motivation of this section comes from the question of Impagliazzo et al. [2022] on whether the class of parities can be PAC learned by an efficient replicable algorithm when the marginal distribution $\mathcal{D}$ is not uniform over $\{0,1\}^d$. Let us define our concept class of interest.[10]

**Definition 4.1** (Affine Parities). *Let $S \subseteq [d]$ be some subset of $[d]$ and $b \in \{0,1\}$ a bias term. Define $f_{S,b} : \{0,1\}^d \to \{0,1\}$ to be the biased parity of the bits in $S$, namely $f_{S,b}(x) = b + \sum_{i \in S} x_i$. The concept class of affine parities over $\{0,1\}^d$ is the set $\mathscr{C} = \{f_{S,b} : S \subseteq [d], b \in \{0,1\}\}$. For any distribution $\mathcal{D}$ over $\{0,1\}^d$, we consider the supervised learning problem $\mathsf{AffParity}$, where the learner observes i.i.d. samples of the form $(x, f^\star(x))$, where $x \sim \mathcal{D}$ and $f^\star \in \mathscr{C}$.*

As observed by Impagliazzo et al. [2022], there is a replicable algorithm for PAC learning parities (i.e., the subclass Parity obtained by setting $b = 0$ in AffParity) under the uniform distribution: draw roughly $O(d)$ samples so that with high probability the dataset will contain a basis. Then, in two distinct executions, the sets that standard Gaussian elimination outputs will be the same. A similar approach works for the class of affine parities and is described below.

---

[9]By performing an iterated doubling if necessary, we may assume without loss of generality that $\ell$ is known.
[10]We consider a superset of parities which we call *affine parities* as we would like our concept class to be closed under restrictions (cf. Definition 3.1).

**Lemma 4.2.** *The concept class of affine parities* AffParity *over* $\{0,1\}^d$ *admits a $\rho$-replicable algorithm that perfectly learns any concept with respect to the uniform distribution $\mathcal{U}$ with probability of success at least $1 - \beta$. The algorithm has $O(\mathrm{poly}(d, \log 1/\rho\beta))$ sample and time complexity.*

The algorithm that attains the guarantees of Lemma 4.2 is a simple adaptation of the Gaussian elimination for learning standard parities. We provide the pseudocode in Algorithm C.1 and defer the proof of correctness to Appendix C.1.

**Application of Theorem 3.2**   Since the class of affine parities over $\{0,1\}^d$ is closed under restrictions (cf. Lemma C.8) and there is a learner for this class under uniform marginals that is replicable and efficient (cf. Lemma 4.2), we can obtain an algorithm that replicably PAC learns the class AffParity under more general distributions. In particular, the following result is an immediate application of our lifting framework.

**Corollary 4.3.** *Let $\alpha, \rho \in (0,1)$ and $\beta \in (0, \rho/3)$. Let $\mathcal{X} = \{0,1\}^d$. For $M = \mathrm{poly}(d, 1/\alpha, 1/\rho, 1/\beta)^{O(\ell)}$, the following cases hold:*
*(a) For any (unknown) monotone distribution $\mathcal{D}$ over $\mathcal{X}$ with decision tree complexity $\ell$, there exists an algorithm that is a $\rho$-replicable learner for the concept class* AffParity *under $\mathcal{D}$, and requires $M$ samples and running time $\mathrm{poly}(M)$ to get accuracy $\alpha$ and confidence $\beta$.*
*(a) For any distribution $\mathcal{D}$ over $\mathcal{X}$ with decision tree complexity $\ell$, there exists an algorithm that is a $\rho$-replicable learner for the concept class* AffParity *under $\mathcal{D}$, and requires $M$ labeled examples, $M$ conditional samples from $\mathcal{D}$ and running time $\mathrm{poly}(M)$ to get accuracy $\alpha$ and confidence $\beta$.*

**Back to the Question of Impagliazzo et al. [2022].**   Impagliazzo et al. [2022] raised the question of when parities over $\{0,1\}^d$ can be efficiently PAC learned by a replicable algorithm under some marginal distribution $\mathcal{D}$ that is not uniform or, more broadly, that causes Gaussian elimination to be non-replicable. Our Corollary 4.3 makes progress on this question. First, we note that in our setting, we do not require knowledge of $\mathcal{D}$. Second, one can design examples of (monotone) distributions for which our lifting framework produces a replicable polynomial time PAC learner but naive Gaussian elimination[11] fails to be replicable with constant probability (cf. Theorem C.5). Note that even PAC learning parities for the distribution described in Theorem C.5 is SQ-hard. We believe that our lifting framework can be seen as a systematic way of bypassing some instabilities arising from the algebraic structure of standard Gaussian elimination.

As a final remark, we note that one could potentially design much more complicated (still monotone) distributions than the one of Theorem C.5, where even various adaptations of Gaussian elimination (e.g., pre-processing of the dataset, data deletion) would fail to be replicable but our lifting framework would guarantee replicability (and efficiency, provided small decision tree complexity). On the other hand, it is not evident whether parity learning remains SQ-hard under these "harder" distributions.

## 5   Efficient Replicability and Private Learning

In this section, we study connections between efficient DP learnability and efficient replicable learnability of a concept class. As we mentioned in the introduction, Bun et al. [2023] and subsequently Kalavasis et al. [2023] established a *statistical* equivalence between approximate DP learnability and replicable learnability of a concept class when the domain is finite [Bun et al., 2023] or countable [Kalavasis et al., 2023]. Moreover, Bun et al. [2023] showed that this equivalence does not hold when one takes into account the computational complexity of these tasks.

**Proposition 5.1** (Section 4.1 in Bun et al. [2023])**.** *There exists a class that is efficiently PAC learnable by an approximate DP algorithm but, assuming one-way functions exist, it cannot be learned efficiently by a replicable algorithm.*

We remark that there is an efficient converse transformation, i.e., a computationally efficient transformation from a replicable learner to an approximate DP learner [Bun et al., 2023]. Inspired by Gonen et al. [2019], we ask whether one can transform a *pure* DP learner to a replicable one. Our main result, proven in Appendix F, is the following.

**Theorem 5.2** (From Pure DP Learner to Replicable Learner)**.** *Let $\mathcal{X}$ be some input domain, $\mathcal{Y} = \{0,1\}$, and $\mathcal{D}_{\mathcal{X}\mathcal{Y}}$ be a distribution on $\mathcal{X} \times \mathcal{Y}$ that is realizable with respect to some concept class $\mathscr{C}$. Let $\mathcal{A}$ be a pure DP learner that, for any $\alpha, \varepsilon, \beta \in (0,1)$, needs $m(\alpha, \varepsilon, \beta, \mathscr{C}) =$*

---

[11]We note that the examples we design are difficult instances for the standard Gaussian elimination algorithm but some pre-processing of the dataset (e.g., deleting some constant fraction of samples) could potentially make Gaussian elimination replicable.

poly($1/\alpha$, $1/\varepsilon$, $\log(1/\beta)$, $\dim(\mathscr{C})$) *i.i.d. samples from $\mathcal{D}_{\mathcal{X}\mathcal{Y}}$ and* poly($m$) *running time to output a hypothesis that has error at most $\alpha$, with probability $1 - \beta$ in an $\varepsilon$-DP way. Then, for any $\alpha', \rho, \beta' \in (0, 1)$ there is a $\rho$-replicable learner $\mathcal{A}'$ that outputs a hypothesis with error at most $\alpha'$ with probability at least $1 - \beta'$ and requires* poly($1/\alpha'$, $1/\rho$, $\log(1/\beta')$, $\dim(\mathscr{C})$) *i.i.d. samples from $\mathcal{D}_{XY}$ and* poly($1/\alpha'$, $1/\rho$, $\log(1/\beta')$) $\cdot \exp(\dim(\mathscr{C}))$ *running time.*

In the above, we denote by $\dim(\mathscr{C})$ some dimension that describes the complexity of the concept class $\mathscr{C}$ that arises in the sample complexity of our pure DP learner. A natural candidate is the *representation dimension* [Kasiviswanathan et al., 2011]. As we alluded to before, this transformation is efficient with respect to the parameters $\alpha, \beta, \rho$. On the other side, the sample complexity is polynomial in the representation dimension but the running time is exponential. We leave as an open question if it is possible to avoid this dependence.

We remark that in principle, one could use the reduction from Bun et al. [2023]. The catch is that this reduction is based on correlated sampling so it requires i) the output space of the algorithm to be finite and ii) even under finite output spaces, it needs exponential time in the size of that space.

## 6 Conclusion

In this work, we have studied the computational aspects of replicability and several connections to other important notions in learning theory including online learning, SQ learning, and DP PAC learning. We believe that there are several interesting questions left open from our work. First, it would be interesting to see if there is a computationally efficient transformation from online learners to replicable learners. Then, it would be important to derive replicable learners from pure DP learners which are efficient with respect to the complexity of the underlying concept class. Regarding parities, it is still open whether we can design efficient replicable algorithms for *every* distribution $\mathcal{D}$.

## Acknowledgments

Alkis Kalavasis was supported by the Institute for Foundations of Data Science at Yale. Amin Karbasi acknowledges funding in direct support of this work from NSF (IIS-1845032), ONR (N00014- 19-1-2406), and the AI Institute for Learning-Enabled Optimization at Scale (TILOS). Grigoris Velegkas was supported in part by the AI Institute for Learning-Enabled Optimization at Scale (TILOS). Felix Zhou acknowledges the support of the Natural Sciences and Engineering Research Council of Canada (NSERC).

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

# A Notation and Preliminaries

For standard PAC learning definitions, we refer to the book of Kearns and Vazirani [1994].

## A.1 Replicable Learning

Following the pioneering work of Impagliazzo et al. [2022], we consider the definition of a replicable learning algorithm.

**Definition 1.1** (Replicable Algorithm; Impagliazzo et al., 2022). *Let $\mathcal{R}$ be a distribution over random strings. A learning algorithm $\mathcal{A}$ is $n$-sample $\rho$-replicable under distribution $\mathcal{D}$ if for two independent sets $S, S' \sim \mathcal{D}^n$ it holds that $\Pr_{S, S' \sim \mathcal{D}^n, r \sim \mathcal{R}}[\mathcal{A}(S, r) \neq \mathcal{A}(S', r)] \leq \rho$. We will say that $\mathcal{A}$ is replicable if the above holds uniformly over all distributions $\mathcal{D}$.*

In words, $\mathcal{A}$ is replicable if sharing the randomness across two executions on different i.i.d. datasets yields the exact same output with high probability. We can think of $\mathcal{A}$ as a training algorithm which is replicable if by fixing the random seed, it outputs the exact same model with high probability.
One of the most elementary statistical operations we may wish to make replicable is mean estimation. This operation can be phrased more broadly using the language of *statistical queries*.

**Definition A.1** (Statistical Query Oracle; Kearns, 1998). *Let $\mathcal{D}$ be a distribution over the domain $\mathcal{X}$ and $\phi : \mathcal{X} \to \mathbb{R}$ be a statistical query with true value $v^\star := \lim_{n \to \infty} \phi(X_1, \ldots, X_n) \in \mathbb{R}$. Here $X_i \sim_{i.i.d.} \mathcal{D}$ and the convergence is understood in probability or distribution. Let $\alpha, \beta \in (0, 1)^2$. A statistical query (SQ) oracle outputs a value $v$ such that $|v - v^\star| \leq \alpha$ with probability at least $1 - \beta$.*

The SQ framework appears in various learning theory contexts (see e.g., Blum et al. [2003], Gupta et al. [2011], Goel et al. [2020], Fotakis et al. [2021] and the references therein). In the SQ model, the learner interacts with an oracle in the following way: the learner submits a statistical query to the oracle and the oracle returns its true value, after adding some noise to it.
The simplest example of a statistical query is the sample mean $\phi(X_1, \ldots, X_n) = \frac{1}{n} \sum_{i=1}^{n} X_i$. Impagliazzo et al. [2022] designed a replicable SQ-query oracle for sample mean queries with bounded co-domain. Esfandiari et al. [2023b] generalized the result to simultaneously estimate the means of multiple random variables with unbounded co-domain under some regularity conditions on their distributions (cf. Theorem B.4). The idea behind both results is to use a replicable rounding technique introduced in Impagliazzo et al. [2022] which allows one to sacrifice some accuracy of the estimator in exchange for the replicability property.

## A.2 Private Learning

**Differential Privacy.** A foundational notion of algorithmic stability is that of Differential Privacy (DP) [Dwork et al., 2006]. For $a, b, \varepsilon, \delta \in (0, 1)$, let $a \approx_{\varepsilon, \delta} b$ denote the statement $a \leq e^\varepsilon b + \delta$ and $b \leq e^\varepsilon a + \delta$. We say that two probability distributions $P, Q$ are $(\varepsilon, \delta)$-indistinguishable if $P(E) \approx_{\varepsilon, \delta} Q(E)$ for any measurable event $E$.

**Definition A.2** (Approximate DP Algorithm; Kasiviswanathan et al., 2011). *A learning algorithm $A$ is an $n$-sample $(\varepsilon, \delta)$-differentially private if for any pair of samples $S, S' \in (\mathcal{X} \times \{0, 1\})^n$ that disagree on a single example, the induced posterior distributions $A(S)$ and $A(S')$ are $(\varepsilon, \delta)$-indistinguishable.*

In the previous definition, when the parameter $\delta = 0$, we say that the algorithm satisfies (pure) $\varepsilon$-DP. We remind the reader that, in the context of PAC learning, any hypothesis class $\mathcal{C}$ can be PAC-learned by an approximate differentially private algorithm if and only if it has finite *Littlestone dimension*, i.e., there is a qualitative equivalence between online learnability and private PAC learnability [Alon et al., 2019, Bun et al., 2020, Ghazi et al., 2021a, Alon et al., 2022].

## A.3 Online Learning

We consider the no-regret model of online learning. Recall Littlestone's model of (realizable) online learning [Littlestone, 1988] defined via a two-player game between a learner and an adversary. Let $\mathcal{C}$ be a given concept class.
In each round of the game $t = 1, \ldots, T$ where $T$ is a time horizon known to a (randomized) learner, the interaction is the following:

1) The adversary selects some features $x_t \in \{0, 1\}^d$.
2) The learner predicts a label $\hat{b}_t \in \{0, 1\}$, potentially using randomization,
3) The adversary, who observes the distribution of the choice of the learner but not its realization, chooses the correct label $b_t = c(x_t)$ under the constraint that there exists some $c_t^\star \in \mathcal{C}$ such that $c_\tau^\star(x_\tau) = b_\tau$, for all $\tau \leq t$.

The goal of the learner is to minimize its regret, defined by

$$R_T := \max_{h \in \mathscr{C}} \left\{ \sum_{t=1}^{T} \mathbb{1}\{\hat{b}_t \neq c(x_t)\} - \mathbb{1}\{h(x_t) \neq c(x_t)\} \right\}$$

$$= \sum_{t=1}^{T} \mathbb{1}\{\hat{b}_t \neq c(x_t)\} - 0 \qquad\qquad h = c$$

$$= \sum_{t=1}^{T} \mathbb{1}\{\hat{b}_t \neq c(x_t)\}.$$

Thus the learner aims to compete with the best concept in hindsight from the class $\mathscr{C}$. However, realizability ensures that the best such concept makes no mistakes so the regret is defined only in terms of the number of mistakes of the learner.

We now introduce the definition of an *efficient no-regret* learning algorithm used by Bun [2020]. In the definition below, we write $|c|$ to denote the length of a minimal description of the concept $c$.

**Definition A.3** (Efficient No-Regret Learning; [Bun, 2020]). *We say that a learner* efficiently no-regret learns $\mathscr{C}$ *if there is some $\eta > 0$ such that for every adversary, it achieves expected regret*

$$\mathbb{E}[R_T] = \mathrm{poly}(d, |c|)T^{1-\eta}$$

*using time $\mathrm{poly}(d, |c|, T)$ in every round.*

There are two non-standard features of this definition. First, no-regret algorithms are typically only required to achieve sublinear regret $o(T)$ in $T$, whereas we require it to be strongly sublinear $T^{1-\eta}$. A stronger condition like this is needed to make the definition nontrivial since the sample space is finite. Indeed, suppose $T = 2^d$, then the trivial random guessing algorithm attains a regret bound of

$$\frac{T}{2} = \frac{dT}{2d}$$
$$= \frac{dT}{2\log T}$$
$$= \mathrm{poly}(d)o(T).$$

Many no-regret algorithms such as the multiplicative weights update algorithm achieve strongly sublinear regret.

Second, it would be more natural to require the learner to run in $\mathrm{poly}(\log T)$ time, the description length of the time horizon, rather than the value of $T$ itself. The relaxed formulation only makes the separation stronger.

## B  Efficient Replicability and Online Learning

### B.1  One-Way Sequences

Our exposition follows that of Bun [2020]. For every dimension $d \in \mathbb{N}$, Blum [1994] defines a concept class $\mathcal{OWS}_d$ consisting of functions over the domain $\{0,1\}^d$ that can be represented using $\mathrm{poly}(d)$ bits and evaluated in $\mathrm{poly}(d)$ time. The concepts of $\mathcal{OWS}_d$ are indexed by bit strings $s \in \{0,1\}^k$, where $k = \lfloor \sqrt{d} \rfloor - 1$

$$\mathcal{OWS}_d = \left\{ c_s : s \in \{0,1\}^k \right\}.$$

We will usually omit the index $d$ when it is clear from the context. Each $c_s : \{0,1\}^d \to \{0,1\}$ is defined using two efficiently representable and computable functions

$$G : \{0,1\}^k \times \{0,1\}^k \to \{0,1\}^{d-k},$$
$$f : \{0,1\}^k \times \{0,1\}^k \to \{0,1\}$$

that are based on the Goldreich-Goldwasser-Micali pseudorandom function family [Goldreich et al., 1986]. We omit the definition of these functions since it does not impede us towards our goal and refer the reader to Blum [1994] for details. Intuitively, $G(i,s)$ computes the string $\sigma_i$ described in the introduction and $f(i,s)$ computes its label $b_i$. We can think of $s$ as the random seed which is generated from some source of true randomness which is then used to construct $G, f$.

For convenience, we identify $\{0,1\}^k \equiv [2^k]$. Then $c_s$ is defined as

$$c_s(i, \sigma) = \begin{cases} 1, & G(i, s) = \sigma, f(i, s) = 1, \\ 0, & \text{else}. \end{cases}$$

We see that $c_s(i, \sigma)$ encodes both the string $\sigma_i$ as well as its label $b_i$, for every random seed $s \in \{0,1\}^k$.

The two relevant properties of the strings $\sigma_i$ are summarized below.

**Proposition B.1** (Forward is Easy; Blum, 1994)**.** *There is an efficiently computable function*

$$\texttt{ComputeForward} : \{0,1\}^k \times \{0,1\}^k \times \{0,1\}^{d-k} \to \{0,1\}^{d-k} \times \{0,1\}$$

*such that for every $j > i$,*

$$\texttt{ComputeForward}(j, i, G(i, s)) = (G(j, s), f(j, s)).$$

**Proposition B.2** (Reverse is Hard; Blum, 1994)**.** *Assuming the existence of one-way functions, there exist functions $G : \{0,1\}^k \times \{0,1\}^k \to \{0,1\}^{d-k}$ and $f : \{0,1\}^k \times \{0,1\}^k \to \{0,1\}$ satisfying the following. Let $\mathcal{O}$ be an oracle that on input $(j, i, G(i, s))$, outputs $(G(j, s), f(j, s))$ for any $j > i$. Let $A$ be any polynomial time randomized algorithm and let $A^{\mathcal{O}}$ denote the algorithm with access to the oracle $\mathcal{O}$. For every $i \in \{0,1\}^k$,*

$$\Pr\left[A^{\mathcal{O}}(i, G(i, s)) = f(i, s)\right] \leq \frac{1}{2} + \mathrm{negl}(d),$$

*where the probability is taken over the internal randomness of $A$ and uniformly random $s \sim \mathcal{U}\left(\{0,1\}^k\right)$.*

The above proposition states that no efficient algorithm can outperform random guessing when trying determinew the label of a string, even given access to a "compute-forward" oracle.

## B.2   Hardness of Efficiently Online Learning $\mathcal{OWS}$

Blum [1994] used Proposition B.2 to show that $\mathcal{OWS}$ cannot be learned in the *mistake bound* model, a more stringent model of online learning compared to the no-regret setting. Later, Bun [2020] adapted the argument to the no-regret setting, making the separation stronger.

**Theorem B.3** ([Bun, 2020])**.** *Assuming the existence of one-way functions, $\mathcal{OWS}$ cannot be learned by an efficient no-regret algorithm.*

Let $|c|$ denote the length of a minimal description of the concept $c$. Recall that a learner *efficiently no-regret learns a class $\mathscr{C}$* if there exists $\eta > 0$ such that for every adversary and any $c \in \mathscr{C}$, it achieves $\mathbb{E}[R_T] = \mathrm{poly}(d, |c|) \cdot T^{1-\eta}$ (strongly sublinear) using time $\mathrm{poly}(d, |c|, T)$ in every round. As mentioned in Appendix A.3, this requirement of strongly sublinear regret is necessary to make the definition non-trivial as the random guessing algorithm attains $o(T)$ regret in finite domains.

## B.3   Replicable Query Rounding

Impagliazzo et al. [2022] designed replicable statistical query oracles (cf. Definition A.1) for bounded co-domains and Esfandiari et al. [2023b] generalized their results to multiple general queries with unbounded co-domain (cf. Theorem B.4), assuming some regularity conditions on the queries.

We illustrate the idea behind the rounding procedure applied to the task of mean estimation. An initial observation is that across two executions, the empirical mean concentrates about the true mean with high probability. Next, we discretize the real line into equal-length intervals with a random shift. It can be shown that both points fall into the same random interval with high probability and outputting the midpoint of said interval yields the replicability guarantee.

For completeness, we include the pseudocode and a proof of the replicable rounding procedure.

**Theorem B.4** (Replicable Rounding; Impagliazzo et al., 2022, Esfandiari et al., 2023b)**.** *Let $\mathcal{D}$ be a distribution over some domain $\mathcal{X}$. Let $\alpha, \rho \in (0, 1)$ and $\beta \in (0, \rho/3)$. Suppose we have a sequence of statistical queries $g_1, \ldots, g_T : \mathcal{X} \to \mathbb{R}$ with true values $\mu_1, \ldots, \mu_T$ and sampling $n$ independent points from $\mathcal{D}$ ensures that*

$$\max_{t \in [T]} |g_t(x_1, \ldots, x_n) - \mu_t| \leq \alpha$$

*with probability at least $1 - \beta$.*

*Then there is a polynomial-time time postprocessing procedure* rRound *(cf. Algorithm B.1) such that each composition* rRound$(g_t, \alpha, \rho)$ *is $\rho$-replicable. Moreover, the outputs $\hat{\mu}_t =$* rRound$(g_t(x_1, \ldots, x_n), \alpha, \rho)$ *satisfy*

$$\max_{t \in [T]} |\hat{\mu}_t - \mu_t| \leq \frac{4\alpha}{\rho}$$

*with probability at least $1 - \beta$.*

Note that if we wish for the sequence of rounding steps to be $\rho$-replicable overall, we can simply run each rounding step with parameter $\rho/T$.

---

**Algorithm B.1** Replicable Rounding

---

1: rRound(query values $g_1, \ldots, g_T$, accuracy $\alpha$, replicability $\rho$):
2: $L \leftarrow 6\alpha/\rho$
3: Sample $L_0 \sim U[0, L]$
4: Discretize the real line $\ldots, [L_0 - L, L_0), [L_0, L_0 + L), [L_0 + L, L_0 + 2L), \ldots$ into disjoint intervals
5: **for** $t = 1, \ldots, T$ **do**
6:     Round $g_t$ to the midpoint of the interval, $\hat{\mu}_t$
7: **end for**
8: **return** $\hat{\mu} \in \mathbb{R}^T$.

---

*Proof (Theorem B.4).* Discretize the real line as disjoint intervals.

$$\ldots, [-L, 0), [0, L), [L, 2L), \ldots.$$

We will choose the value of $L$ later. Consider adding a uniformly random offset $L_0 \sim U[0, L]$ so the discretization becomes

$$\ldots, [L_0 - L, L_0), [L_0, L_0 + L), [L_0 + L, L_0 + 2L), \ldots.$$

For each $t \in T$, we round the estimate $g_t(x_{1:n}) := g_t(x_1, \ldots, x_n)$ to the midpoint of the interval it falls into. Let $\hat{\mu}_t$ be the rounded estimate that we output. From hereonforth, we condition on the event $\max_t |g_t(x_{1:n}) - \mu_t| \leq \alpha$.

**Replicability of Algorithm B.1.** Fix $t \in T$. Consider the output across two runs $\hat{\mu}_t, \hat{\mu}'_t$. As long as the raw estimates $g_t(x_{1:n}), g_t(x'_{1:n})$ fall in the same interval, the outputs will be exactly the same. This occurs with probability

$$\frac{|g_t(x_{1:n}) - g_t(x'_{1:n})|}{L} \leq \frac{2\alpha}{L}.$$

Choosing $L = 6\alpha/\rho$ ensures this value is at most $\rho/3$. Accounting for the $2\beta \leq 2\rho/3$ probability of the estimates $g_t(x_{1:n})$ failing to concentrate, the output $\hat{\mu}_t$ is $\rho$-replicable.

**Correctness of Algorithm B.1.** The rounding incurs an additive error of at most $L/2$. The choice of $L = 6\alpha/\rho$ means the total error is at most

$$\alpha + \frac{3\alpha}{\rho} \leq \frac{4\alpha}{\rho}.$$

$\square$

## B.4 Replicable Quantile Estimation

In this section we provide an algorithm that replicably estimates *quantiles* of a distribution. This will be useful in providing the computational separation between replicable PAC learning and online learning. We believe that it can have applications beyond the scope of our work.

We first present a well-known concentration inequality regarding CDFs of random variables due to Dvoretzky, Kiefer, and Wolfowitz.

**Theorem B.5** (Dvoretzky–Kiefer–Wolfowitz Inequality; Dvoretzky et al., 1956, Massart, 1990)**.** *Let $X_1, \ldots, X_n$ be i.i.d. random variables with CDF $F$. Let $F_n$ denote the empirical distribution function given by $F_n(x) := \frac{1}{n} \sum_{i=1}^n \mathbb{1}\{X_i \leq x\}$. For any $\alpha > 0$,*

$$\Pr\left[\sup_{x \in \mathbb{R}} |F_n(x) - F(x)| > \alpha\right] \leq 2\exp(-2n\alpha^2).$$

In other words, we require at most $n = (1/2\alpha^2)\ln(2/\beta)$ samples to ensure that with probability at least $1 - \beta$, the empirical CDF uniformly estimates the true CDF with error at most $\alpha$.

The replicable quantile estimation algorithm for discrete and bounded distributions can be found in Algorithm B.2. The high-level idea is to perform a (replicable) binary search over the support of the distribution and to check whether the empirical CDF evaluated at some point $x$ is above or below the target quantile $q$.

---

**Algorithm B.2** Replicable Quantile Estimation

---

1: `rQuantileEst`(samples $x_1, \ldots, x_n$, quantile $q$, accuracy $\alpha$, replicability $\rho$, confidence $\beta$):
2: $F_n(i) \leftarrow \sum_{j=1}^{n} \mathbb{1}\{x_j \leq i\}$ // Implicitly for all $i \in [R]$
3: $\ell \leftarrow 0; h \leftarrow R$
4: **while** $\ell < h - 1$ **do**
5:     $m \leftarrow (\ell + h)/2$
6:     $\widetilde{F}_n(m) \leftarrow \text{rRound}(F_n(m), \alpha\rho/4\log_2(R), \rho/\log_2(R))$
7:     **if** $\widetilde{F}_n(m) \geq q$ **then**
8:         $h \leftarrow m$
9:     **else**
10:        $\ell \leftarrow m$
11:     **end if**
12: **end while**
13: **return** $h$.

---

**Theorem B.6** (Replicable Quantile Estimation). *Let $\alpha, \rho \in (0, 1)$ and $\beta \in (0, \rho/3)$. Suppose we have access to*

$$m = \frac{16 \log_2^2(R)}{2\alpha^2\rho^2}\ln\frac{2}{\beta} = O\left(\frac{\log^2 R}{\alpha^2\rho^2}\ln\frac{1}{\beta}\right)$$

*i.i.d. samples from some distribution over $[R]$ with CDF $F$ and $q \in [0, 1]$ is the desired quantile level. Algorithm B.2 terminates in $O(\text{poly}(n))$ time, is $\rho$-replicable, and with probability at least $1 - \beta$, outputs some $x \in [R]$ such that*

$$F(x) \geq q - \alpha, \qquad F(x - 1) < q + \alpha.$$

Our replicable quantile estimation differs from the replicable median algorithm of [Impagliazzo et al., 2022] in at least two ways. Firstly, the replicable median algorithm of [Impagliazzo et al., 2022] seems to rely heavily on properties of (approximate) medians in order to satisfy the approximation guarantees. On the other hand, our algorithm works regardless of the desired quantile. Secondly, their median algorithm relies on a non-trivial recursive procedure while our replicable quantile algorithm is considerably simpler and is based on a concentration of the CDF through the DKW inequality.

*Proof (Theorem B.6).* By the DKW inequality (Theorem B.5), sampling

$$m = \frac{1}{2(\alpha\rho/4\log_2(R))^2}\ln\frac{2}{\beta}$$

points from our distribution ensures that

$$\sup_{x \in [R]} |F_n(x) - F(x)| \leq \frac{\alpha\rho}{4\log_2(R)}$$

with probability at least $1 - \beta$. The proof is divided into two steps. We first show the correctness of our algorithm and then argue about its replicability.

**Correctness of Algorithm B.2.** The loop invariant we wish to maintain is that

$$F(h) \geq q - \alpha, \qquad F(\ell) < q + \alpha.$$

This is certainly initially true since our distribution is over $[R] = \{1, \ldots, R\}$ and hence there is no mass at or below $\ell = 0$ while all the mass is at or below $h = R$.

Let $m_1, \ldots, m_T$ denote the midpoints chosen by binary search with $T = \log_2(R)$. By Theorem B.4, if we condition on the success of all executions of $\text{rRound}(F_n(m_t), \alpha\rho/4\log_2(R), \rho/\log_2(R))$, then their outputs $\widetilde{F}_n(m_t)$ estimate $F(m_t)$ with additive error at most

$$\frac{4\alpha\rho/4\log_2(R)}{\rho/\log_2(R)} = \alpha.$$

We update $h \leftarrow m_t$ only if

$$q \leq \widetilde{F}_n(m_t) \leq F(m_t) + \alpha \implies q - \alpha \leq F(m_t).$$

Similarly, we update $\ell \leftarrow m_t$ only if

$$q > \widetilde{F}_n(m_t) \geq F(m_t) - \alpha \implies q + \alpha > F(m_t).$$

Hence the loop invariant is maintained.
At termination, $\ell = h - 1$ with

$$F(h) \geq q - \alpha$$
$$F(h-1) < q + \alpha$$

by the loop invariant as desired.

**Replicability of Algorithm B.2.** By Theorem B.4, assuming all prior branching decisions are identical, each new branching decision in Algorithm B.2 is $(\rho/\log_2(R))$-replicable. Since we make at most $\log_2(R)$ decisions, the entire algorithm is $\rho$-replicable. $\qquad\square$

### B.5 An Efficient Replicable Learner for $\mathcal{OWS}_d$

Equipped with the replicable quantile estimator from Theorem B.6, we are ready to present our efficient replicable $\mathcal{OWS}_d$ PAC learner. We outline below a high-level description of our algorithm which can be found in Algorithm B.3.

1) We first replicably estimate the mass of all the elements that have positive labels. If it is much smaller than some threshold $O(\alpha)$, then we output the zero hypothesis.
2) Then we take sufficiently many samples to get enough data points with positive label and we run the replicable quantile estimation on the marginal distribution of features with positive label to get some $x \in [2^k]$.
3) Next, we take sufficiently many samples to get a positive point at or below $x$ and then we forward compute the label of $x$.
4) The hypothesis we return is the following: If its input $(i, \sigma)$ has index less than $i^*$, it outputs 0. If its input is exactly $(i^*, \sigma^*)$, it outputs the previously computed label. Otherwise, it forward computes the label using $i, \sigma$.

---

**Algorithm B.3** Replicable Learner for $\mathcal{OWS}_d$

1: rLearnerOWSd(samples $(i_1, \sigma_1), \ldots, (i_n, \sigma_m)$, accuracy $\alpha$, replicability $\rho$, confidence $\beta$):
2: Let $S_+ := (i_{j_1}, \sigma_{j_1}), \ldots, (i_{j_n}, \sigma_{j_n})$ be the subsequence of positive samples, where $i_{j_k} \leq i_{j_{k+1}}$.
3: $\widehat{p} \leftarrow$ rRound$(n/m, \rho\alpha/48, \rho/3)$
4: **if** $\widehat{p} < \alpha/2$ **then**
5:     **return** All-zero hypothesis.
6: **end if**
7: $i^* \leftarrow$ rQuantileEst$(\{i_{j_1}, \ldots, i_{j_n}\}, \alpha/2, \alpha/4, \rho/3, \beta/3)$
8: **if** $i_{j_1} \geq i^*$ **then**
9:     **return** "FAILURE"
10: **end if**
11: $(\sigma^*, b^*) \leftarrow$ ComputeForward$(i^*, i_{j_1}, \sigma_{j_1})$
12: **return** hypothesis $h(i, \sigma) :=$ "
    If $i < i^*$, output 0.
    If $i = i^*$, output $b^*$ if $\sigma^* = \sigma$ and 0 otherwise.
    If $i > i^*$, get $(\hat{\sigma}, \hat{b}) \leftarrow$ ComputeForward$(i, i^*, \sigma^*)$ and output $\hat{b}$ if $\sigma = \hat{\sigma}$ and 0 otherwise."

---

**Theorem B.7.** *Let $\rho, \alpha \in (0, 1)$ and $\beta \in (0, \rho/3)$. Algorithm B.3 is an efficient $\rho$-replicable $(\alpha, \beta)$-PAC learner for $\mathcal{OWS}_d$ with sample complexity*

$$m = \max\left(\frac{392}{\alpha^2\rho^2}\ln\frac{6}{\beta}, \frac{9216k^2}{\alpha^3\rho^2}\ln\frac{6}{\beta}, \frac{32}{\alpha^2}\ln\frac{6}{\beta}\right) = O\left(\frac{d^2}{\alpha^3\rho^2}\ln\frac{1}{\beta}\right)$$

*and time complexity*

$$O(\text{poly}(m)).$$

*Proof (Theorem B.7).* The indicator random variable $I_i := \mathbb{1}\{\sigma_i = 1\}$ is a bounded random variable and its sample mean is precisely $n/m$. Let $p \in [0, 1]$ denote the probability mass of the positively labeled elements. By an Hoeffding inequality, $|n/m - p| \leq \alpha$ with probability at least

$$2 \exp\left(-2m\alpha^2\right).$$

Since we have

$$m \geq \frac{392}{\alpha^2 \rho^2} \ln \frac{6}{\beta},$$

then $|m/n - p| \leq \rho\alpha/48$ with probability at least $\beta/6$. By Theorem B.4, the rounded estimate $\widehat{p}$ is $\rho/3$-replicable and satisfies

$$|\widehat{p} - p| \leq \alpha/4$$

with probability at least $\beta/6$. From hereonforth, we condition on the success of this event.

**Correctness of Algorithm B.3.** First suppose $\widehat{p} < \alpha/2$. Then

$$p \leq \widehat{p} + \alpha/4 < \alpha.$$

Then Algorithm B.3 always returns the all-zero hypothesis. In this case, the returned hypothesis only makes mistakes on at most $\alpha$ fraction of the population and we are content.
Otherwise, suppose that $\widehat{p} \geq \alpha/2$. Thus

$$p \geq \widehat{p} - \frac{\alpha}{4} \geq \frac{\alpha}{4}$$
$$\frac{n}{m} \geq p - \frac{\rho\alpha}{16} \geq \frac{\alpha}{8}$$
$$n \geq \frac{\alpha}{8}m$$
$$= \frac{1152k^2}{\alpha^2\rho^2} \ln \frac{6}{\beta}.$$

Let $F$ denote the CDF of the conditional distribution over the positively labeled samples. By Theorem B.6, the output of `rQuantileEst` is an index $i^* \in [2^k]$ such that with probability at least $\beta/6$,

$$F(i^*) \geq \frac{\alpha}{2} - \frac{\alpha}{4} \geq \frac{\alpha}{4}$$
$$F(i^* - 1) < \frac{\alpha}{2} + \frac{\alpha}{4} \leq \alpha.$$

We proceed conditioning on the success of the call to `rQuantileEst`.
Since $F(i^*) \geq \alpha/8$, the probability that $i_{j_1} > i^*$ is at most

$$\left(1 - \frac{\alpha}{8}\right)^n \leq e^{-\frac{n\alpha}{8}} \leq \beta/6.$$

The last inequality is due to $n \geq \alpha m/8 \geq (4/\alpha)\ln(6/\beta)$. We proceed conditioning on the event $i_{j_1} \leq i^*$.
So either $i_{j_1} = i^*$ and we know its string $\sigma^*$ and label $b^*$, or $i_{j_1} < i^*$ and Proposition B.1 assures that we can obtain the string $\sigma^*$ and label $b^*$ through forward computation

$$(\sigma^*, b^*) \leftarrow \texttt{ComputeForward}(i^*, i_{j_1}, \sigma_{j_1}).$$

Now, consider the hypothesis $h$ we output and its action on an input $(i, \sigma)$: If $i < i^*$, then $h$ always answers 0 so it is incorrect only on the positive labels. But this happens at most $F(i^* - 1) < \alpha$ of the time. If $i \geq i^*$, $h$ is always correct. Hence the total population error is at most $\alpha$ as desired.
We conditioned on three events, each of which has failure probability at most $\beta/6$. Hence the total probability of failure is at most $\beta/2$.

**Replicability of Algorithm B.3.** By Theorem B.4, the output $\widehat{p}$ is $\rho/3$ replicable. By Theorem B.6, the output $i^*$ is $\rho/3$ replicable. From our analysis above, $i_{j_1} < i^*$ with probability at most $\beta/6$ in each of two executions. Thus the total probability of outputting different classifiers is at most

$$\frac{\rho}{3} + \frac{\rho}{3} + \frac{2\beta}{6} \leq \frac{2\rho}{3} + \frac{\rho}{9} \leq \rho.$$

$\square$

### B.6 Proof of Theorem 2.1

We now restate and prove Theorem 2.1.

**Theorem 2.1.** *Let $d \in \mathbb{N}$. The following hold:*
*(a) [Bun, 2020] Assuming the existence of one-way functions, the concept class $\mathcal{OWS}$ with input domain $\{0,1\}^d$ cannot be learned by an efficient no-regret algorithm, i.e., an algorithm that for some $\eta > 0$ achieves expected regret $\mathbb{E}[R_T] \leq \mathrm{poly}(d) \cdot T^{1-\eta}$ using time $\mathrm{poly}(d,T)$ in every iteration.[12]*
*(b) (Theorem B.7) The concept class $\mathcal{OWS}$ with input domain $\{0,1\}^d$ can be learned by a $\rho$-replicable $(\alpha, \beta)$-PAC learner with sample complexity $m = \mathrm{poly}(d, 1/\alpha, 1/\rho, \log(1/\beta))$ and $\mathrm{poly}(m)$ running time.*

*Proof (Theorem 2.1).* Combining the hardness of efficiently online learning $\mathcal{OWS}$ (cf. Theorem B.3) with the efficient replicable learner of Theorem B.7 completes the the proof of Theorem 2.1. $\square$

## C Efficient Replicability and SQ Learning: Parities

### C.1 The Proof of Lemma 4.2

---

**Algorithm C.1** Replicable Learner for Affine Parities $f(x) = w^\top x + b$ under the Uniform Distribution

---

1: `rAffParity`(accuracy $\alpha$, replicability $\rho$, confidence $\beta$):
2: Draw a single sample $(x^{(0)}, f(x^{(0)}))$ where $x_0 \sim \mathcal{U}$.
3: Draw $O(d \log(1/\rho\beta))$ samples $(x, f(x))$ so that with probability at least $1 - \beta$, we obtain $d$ linearly independent offsets $(z, y) := (x + x^{(0)}, f(x) + f(x^{(0)}))$, say $(z^{(1)}, y^{(1)}), \ldots, (z^{(d)}, y^{(d)})$.
4: Run Gaussian elimination to obtain the unique solution $w$ such that $w^\top z_i = y_i$ for each $i \in [d]$.
5: Compute $b = f(x^{(0)}) + w^\top x^{(0)}$.
6: Return $(w, b)$.

---

We now repeat and prove Lemma 4.2, which states the correctness of Algorithm C.1.

**Lemma 4.2.** *The concept class of affine parities* AffParity *over $\{0,1\}^d$ admits a $\rho$-replicable algorithm that perfectly learns any concept with respect to the uniform distribution $\mathcal{U}$ with probability of success at least $1 - \beta$. The algorithm has $O(\mathrm{poly}(d, \log 1/\rho\beta))$ sample and time complexity.*

*Proof (Lemma 4.2).* For any $i \in [d]$, observe that

$$y^{(i)} := f(x^{(i)}) + f(x^{(0)}) := w^\top x^{(i)} + b + w^\top x^{(0)} + b = w^\top (x^{(i)} + x^{(0)}) =: w^\top z^{(i)}$$

Thus the dataset of offsets $\{(z^{(i)}, y^{(i)})\}$ uniquely determines the linear function $w$. Having learned $w$, we can recover the value of $b$ by evaluating at any point in the original dataset, say at the first point. $\square$

### C.2 Background on Parities and SQ

**Statistical Queries** We start with some background on the SQ model.

**Definition C.1** (SQ Learning). *A concept class $\mathcal{C}$ with input space $\{0,1\}^d$ is learnable from statistical queries with respect to distribution $\mathcal{D}$ if there is a learning algorithm $\mathcal{A}$ such that for any $c \in \mathcal{C}$ and any $\alpha > 0$, $\mathcal{A}$ produces an $\alpha$-approximation of $c$ from statistical queries; furthermore, the running time, the number of queries asked, and the inverse of the smallest tolerance used must be polynomial in $d$ and $1/\alpha$.*

**Definition C.2** (SQ Hardness). *Consider a concept class $\mathcal{C}$ with input space $\{0,1\}^d$. Fix a tolerance $\tau = \mathrm{poly}(1/d)$ and accuracy $\alpha = O(1)$. We say that $\mathcal{C}$ is SQ-hard under distribution $\mathcal{D}$ if any SQ algorithm requires $\omega(\mathrm{poly}(d))$ queries of tolerance at least $\tau$ to $\alpha$-learn $\mathcal{C}$.*

The standard way to show SQ hardness is by showing lower bounds on the so-called *SQ dimension* [Feldman et al., 2017] of the corresponding problems (we omit the formal definition of SQ dimension as it is beyond the scope of the present work). Such lower bounds on this dimension establish lower bounds on the running time of any SQ algorithm for the problem – not on its sample complexity.

---

[12]We remark that this stronger requirement in regret is necessary since the trivial random guessing algorithm achieves $o(T)$ regret in finite domains (cf. Appendix A.3).

**PAC and SQ Learning Parities** The standard class of parities Parity is given by the subset $\{f_{S,0} : S \subseteq [d]\}$ of AffParity. The standard algorithm for learning parity functions works by viewing a set of $n$ labelled examples as a set of $n$ linear equations over the finite field with two elements $\mathbb{F}_2$. Then, Gaussian elimination is used to solve the system and thus find a consistent parity function. This algorithm is extremely brittle to noise: even for the uniform marginal distribution $\mathcal{D} = \mathcal{U}$, the problem of learning parities over $\{0,1\}^d$ does not belong to SQ. In particular, a standard result is the following:

**Fact C.3** ([Kearns, 1998]). *Even learning parities* Parity $\subset$ AffParity *with input space* $\{0,1\}^d$ *is SQ-hard under the uniform distribution* $\mathcal{U}$.

### C.3 Gaussian Elimination is not Replicable

In this section, we give a simple example where Gaussian elimination fails to be replicable.

**Proposition C.4.** *For any $d \geq 3$ and $S \subseteq [d]$ with $2 \notin S \subseteq [d]$, the following holds: There is some $n_d \geq 1$, such that for any $n \geq n_d$, there exists a distribution $\mathcal{D}$ such that the $n$-sample Gaussian elimination algorithm for PAC-learning parities $f(x) = \sum_{i \in S} x_i$ in the realizable setting fails to be $1/17$-replicable.*

*Proof (Proposition C.4).* Let $e_1, \ldots, e_d$ be the standard basis of $\mathbb{Z}_2^d$, $p \in (0, 2/3]$, and consider the distribution $\mathcal{D}$ such that

$$\mathcal{D}(e_1) = p, \qquad \mathcal{D}(e_2) = \frac{2}{3} - p, \qquad \mathcal{D}(e_i) = \frac{1}{3(d-2)}, i \geq 3.$$

The probability of not observing any $e_1$'s after $n$ samples is $(1-p)^n$. We set $p := 1 - \sqrt[n]{1/2} \leq 1/2$ so that $(1-p)^n = 1/2$.

For sufficiently large $n_d \in \mathbb{N}$, we observe all $e_i, i \geq 3$ with probability at least $3/4$ after $n$ samples. Thus with probability at least $\frac{1}{4}$, we observe $e_1, e_3, \ldots, e_d$, and with probability at least $1/4$, we observe $e_3, \ldots, e_d$ but not $e_1$. In the first case, we fully recover $S$ since we are promised that $2 \notin S$. In the second case, our algorithm is unable to determine if $1 \in S$ since we do not observe $e_1$ and the best it can do is randomly guess. Thus the probability we output different classifiers is at least $2 \cdot 1/4 \cdot 1/4 \cdot 1/2 = 1/16$. $\qquad\square$

Proposition C.4 shows that Gaussian elimination fails in general to be replicable regardless of the number of samples requested.

### C.4 Replicably Learning Affine Parities Beyond the Uniform Distribution

We restate our main result.

**Theorem C.5.** *For any dimension $d$, accuracy $\alpha$, confidence $\beta$ and replicability $\rho$, there exists some $n = \text{poly}(d, 1/\alpha, 1/\rho, \log(1/\beta))$ such that there exists a monotone distribution $\mathcal{D}$ over $\{0,1\}^d$ which*

- *(a) Gaussian elimination is not $n$-sample replicable with probability $\Omega(1)$ under $\mathcal{D}$ for the class* AffParity,
- *(b) there exists an $n$-sample $\rho$-replicable algorithm for $(\alpha, \beta)$-PAC learning the class* AffParity *with respect to $\mathcal{D}$ with $\text{poly}(n)$ runtime, and*
- *(c) even learning the class* Parity *is SQ-hard under the distribution $\mathcal{D}$.*

Before proving Theorem C.5, we derive three useful lemmas. Fix a dataset size $n$ to be defined later. We pick the distribution $\mathcal{D}$ to be a Boolean product distribution $(p_1, ..., p_d) \in [0,1]^d$ such that the first $d-1$ coordinates are unbiased ($p_i = 1/2$) for $i \in [d-1]$ and the last coordinate to be highly biased towards 1. In particular, pick $p_d = \mathcal{D}_d(\{1\}) = \sqrt[n]{1/2} \geq 1/2$. This distribution is monotone by construction.

**Lemma C.6.** *The naive $n$-sample Gaussian elimination algorithm fails to be replicable with constant probability over the distribution $\mathcal{D} = (p_1, ..., p_d)$.*

*Proof.* Consider two independent draws $S_1$ and $S_2$ from $\mathcal{D}$ of size $n$. With constant probability $1/2 \cdot 1/2 = 1/4$, $S_1$ will contain vectors that have only 1's in the last coordinate while $S_2$ will contain at least one vector with a zero in the last direction. Then there are exactly two hypotheses that satisfy $S_1$, one that contains the last coordinate and one that does not. This can be seen by reducing to an instance of the $(d-1)$-dimensional parities problem obtained by ignoring the last coordinate and flipping the bits of the labels.

On the other hand, for $n$ sufficiently large, the subset of vectors $S_2^{(1)} \subseteq S_2$ that have 1's in the last direction is satisfied by the same hypotheses as $S_1$. However, only one of the two hypotheses also satisfies the other subset $S_2^{(0)} \subseteq S_2$ consisting of entries with a 0 in the last direction.

Thus the candidate hypotheses that satisfy $S_1, S_2$ differ and the algorithm can at best output a random guess for $S_1$, which will be inconsistent with the output on $S_2$ with overall probability at least $1/2 \cdot 1/4 = 1/8$. $\qquad \square$

**Lemma C.7.** *The decision tree complexity of $\mathcal{D}$ is $\Theta(1)$.*

*Proof.* Since $\mathcal{D}$ is a product distribution, the pmf only takes on 2 different values and thus has depth $\Theta(1)$. $\qquad \square$

**Lemma C.8.** *The class of affine parities* AffParity *with input space $\{0,1\}^d$ is closed under restriction.*

*Proof.* Let $f$ be an arbitrary affine parity function, i.e., $f(x) = b + \sum_{i \in S} x_i$ for some $b \in \{0,1\}$ and $S \subseteq [d]$. We remark that the operator $+$ is in $\mathbb{Z}_2$. Consider an arbitrary direction $i \in [d]$ and arbitrary bit $b' \in \{0,1\}$. We will verify that $f_{i=b'}$ remains in the class of affine parities. If $i \notin S$, then $f_{i=b'} = f \in$ AffParity. Now, if $i \in S$ and $c = 0$, it holds that $f_{i=b'}(x) = b + \sum_{i \in S \setminus \{i\}} x_i \in$ AffParity and if $b' = 1$, it holds that $f_{i=b'}(x) = (b+1) + \sum_{i \in S \setminus \{i\}} x_i \in$ AffParity. Hence, affine parities are closed under restriction. $\qquad \square$

We are now ready to prove Theorem C.5.

*Proof (Theorem C.5).*

**(a)** By Lemma C.6.

**(b)** We can employ the lifting algorithm of Theorem 3.2 to replicably and efficiently learn affine parities since (i) the decision tree complexity is constant, (ii) the distribution is monotone, (iii) affine parities are closed under restriction and (iv) we have an efficient replicable algorithm for affine parities with respect to the uniform distribution (cf. Lemma 4.2).

**(c)** If we consider the subclass $\mathscr{C}' = \{f_{S,0} : S \subseteq [d-1]\}$, we have that $\mathbb{E}_{x \sim \mathcal{D}}[f(x)g(x)] = 0$ for any pair of distinct functions $f, g \in \mathscr{C}'$ due to the product structure and the unbiasedness of the first $d-1$ coordinates of $\mathcal{D}$. This implies the SQ-dimension of the class of parities is at least $2^{d-1} = \Omega(2^d)$ and so learning parities is SQ-hard under $\mathcal{D}$. $\qquad \square$

# D  Lifting Replicable Uniform Learners

In this section, we will provide our general lifting framework for replicable learning algorithms that is needed in order to show Theorem C.5.

## D.1  Preliminaries for Replicable Uniform Lifting

We start this section with some definitions which are necessary to formally state our main result of Theorem 3.2.

**Definition D.1** (Decision Tree (DT)). *A decision tree $T : \{0,1\}^d \to \mathbb{R}$ is a binary tree whose internal nodes query a particular coordinate, and whose leaves are labelled by values. Each instance $x \in \{0,1\}^d$ follows a unique root-to-leaf path in $T$: at any internal node, it follows either the left or right branch depending on the value of the queried coordinate, until a leaf is reached and its value $T(x)$ is returned.*

**Definition D.2** (Decision tree distribution). *We say that a distribution $\mathcal{D}$ over $\{0,1\}^d$ is representable by a depth-$\ell$ decision tree, if its pmf is computable by a depth-$\ell$ decision tree $T$. Specifically, each leaf $t$ is labelled by a value $p_t$, so that $\mathcal{D}(x) = p_t$ for all $x \in t$. This means that the conditional distribution of all points that reach a leaf is uniform.*

Let us consider a distribution $\mathcal{D}$ over $\{0,1\}^d$. The definitions above suggest a natural measure of the complexity of $\mathcal{D}$: The *decision tree complexity*.

**Definition D.3** (Decision Tree Complexity). *The decision tree complexity of a distribution $\mathcal{D}$ over $\{0,1\}^d$ is the smallest integer $\ell$ such that its probability mass function (pmf) can be represented by a depth-$\ell$ decision tree.*

Next, we define a useful structured family of distributions. Let the binary relation $\succeq$ denote pointwise comparison.

**Definition D.4** (Monotone Distribution). *A probability distribution $\mathcal{D}$ over $\{0,1\}^d$ is called monotone if for any $x \succeq y$ it holds that $\mathcal{D}(x) \geq \mathcal{D}(y)$.*

Just as in Blanc et al. [2023], for arbitrary non-monotone probability measures, our algorithm requires access to a *conditional sampling oracle* (cf. Definition D.5) defined below.

**Definition D.5** (Conditional Sampling Oracle; Blanc et al., 2023). *A conditional sampling oracle for a distribution $\mathcal{D}$ over $\{0,1\}^d$ proceeds as follows: suppose we condition on some subset $I$ of the $d$ variables having a fixed value $b \in \{0,1\}^I$. In that case, the oracle generates a sample from the true conditional distribution, i.e., draws a sample $x_{-I} \sim \mathcal{D}(\cdot \mid x_I = b)$.*

**Definition D.6.** *For a tree $T$ and leaves $\ell \in T$, $\mathbb{E}_{\ell \in T} f(\ell) := \sum_{\ell \in T} 2^{-|\ell|} f(\ell)$, where $|\ell|$ is the depth of the path to reach leaf $\ell$.*

As an example of this notation, note that

$$\mathbb{E}_{\ell \in T}\left[\mathbb{E}_{x \sim \mathcal{U}^d}[f(x)|x \in \ell]\right] = \mathbb{E}_{\ell \in T}\left[\sum_{x \in \ell} \frac{f(x)\mathcal{U}(x)}{\mathcal{U}(\ell)}\right],$$

where $\mathcal{U}(\ell)$ is the uniform mass of the leaf $\ell$, i.e., $\mathcal{U}(\ell) = 2^{-n} \cdot 2^{n-|\ell|}$. This implies that $\mathbb{E}_{\ell \in T}[\mathbb{E}_{x \sim \mathcal{U}^d}[f(x)|x \in \ell]] = \mathbb{E}_{x \in \mathcal{U}^d} f(x)$, since the set of leaves partitions the set $\{0,1\}^d$.

## D.2 Main Result: Lifting Replicable Uniform Learners

We now restate our main result Theorem 3.2. For the sake of presentation, we defer its proof to Appendix E.

**Theorem 3.2** (Lifting Replicable Uniform Learners). *Consider a concept class $\mathscr{C}$ of functions $f : \{0,1\}^d \to \{0,1\}$ closed under restrictions. Suppose we are given black-box access to an algorithm such that for any $\alpha', \rho', \beta' \in (0,1)$, given $\mathrm{poly}(d, 1/\alpha', 1/\rho', \log(1/\beta'))$ samples from $\mathcal{U}$,*

*(i) is $\rho'$-replicable with respect to the uniform distribution $\mathcal{U}$,*

*(ii) PAC learns $\mathscr{C}$ under the uniform distribution to accuracy $\alpha'$ and confidence $\beta'$, and,*

*(iii) terminates in time $\mathrm{poly}(d, 1/\alpha', 1/\rho', \log(1/\beta'))$.*

*Let $\alpha, \rho \in (0,1)$ and $\beta \in (0, \rho/3)$. For $m = \mathrm{poly}(d, 1/\alpha, 1/\rho, \log(1/\beta))$ and $M = \mathrm{poly}(d, 1/\alpha, 1/\rho, \log(1/\beta))^{O(\ell)}$, the following cases hold:*

*(a) If $\mathcal{D}$ is a monotone distribution over $\{0,1\}^d$ representable by a depth-$\ell$ decision tree, there is an algorithm that draws $M$ samples from $\mathcal{D}$, is $\rho$-replicable with respect to $\mathcal{D}$, PAC learns $\mathscr{C}$ under $\mathcal{D}$ with accuracy $\alpha$ and confidence $\beta$, and terminates in $\mathrm{poly}(M)$ time.*

*(b) If $\mathcal{D}$ is an arbitrary distribution over $\{0,1\}^d$ representable by a depth-$\ell$ decision tree, there is an algorithm that draws $M$ labeled examples as well as $M$ conditional samples (cf. Definition D.5) from $\mathcal{D}$, is $\rho$-replicable with respect to $\mathcal{D}$, PAC learns $\mathscr{C}$ with respect to $\mathcal{D}$ with accuracy $\alpha$ and confidence $\beta$, and terminates in $\mathrm{poly}(M)$ time.*

We emphasize that our result for monotone distributions $\mathcal{D}$ only requires sample access to $\mathcal{D}$. For arbitrary non-monotone probability measures, our algorithm requires access to a conditional sampling oracle (cf. Definition D.5)

## E Proof of Theorem 3.2

### E.1 Preliminaries

We start this section with some useful definitions, coming from the work of Blanc et al. [2023].

**Definition E.1** (Weighting function of distribution). *Let $\mathcal{D}$ be an arbitrary distribution over $\{0,1\}^d$. We define the weighting function $f_{\mathcal{D}}(x) = 2^d \mathcal{D}(x)$.*

Remark that the scaled pmf satisfies

$$\mathbb{E}_{x \sim \mathcal{U}}[f_{\mathcal{D}}(x)] = \sum_{x \in \{0,1\}^d} 2^{-d} \cdot 2^d \mathcal{D}(x) = 1.$$

For $x \in \{0,1\}^d$ and $i \in [d]$, we write $x^{\sim i}$ to denote the binary vector obtained from $x$ by flipping its $i$-th bit.

**Definition E.2** (Influence of Variables on Distributions). *Let $\mathcal{D}$ be a distribution over $\{0,1\}^d$ and $f_{\mathcal{D}}(x) = 2^d \cdot \mathcal{D}(x)$ be its pmf scaled up by the domain size. The influence of a coordinate $i \in [d]$ on a distribution $\mathcal{D}$ over $\{0,1\}^d$ is the quantity*

$$\mathrm{Infl}_i(f_{\mathcal{D}}) := \mathbb{E}_{x \sim \mathcal{U}}\left[|f_{\mathcal{D}}(x) - f_{\mathcal{D}}(x^{\sim i})|\right]$$

*We also define the* total influence $\mathrm{Infl}(f_{\mathcal{D}}) = \sum_{i \in [d]} \mathrm{Infl}_i(f_{\mathcal{D}})$.

Intuitively, the influence of a variable measures how far that variable is from the uniform distribution. Suppose $f_{\mathcal{D}}$ is computable by a depth-$\ell$ decision tree $T$. Then we can write for any $x \in \{0,1\}^d$

$$\mathcal{D}(x) = \sum_{\ell \in T} p_\ell \cdot \mathbb{1}\{x \in \ell\}.$$

where the sum is taken over the leaves of $T$ and $p_\ell := \mathcal{D}(y)$ is the (same) mass $\mathcal{D}$ assigns to every $y \in \ell$. Then we have

$$
\begin{aligned}
\mathrm{Infl}_i(f_{\mathcal{D}}) &:= \mathbb{E}_{x \sim \mathcal{U}}|f_{\mathcal{D}(x)} - f_{\mathcal{D}}(x^{\sim i})| \\
&= \sum_{x \in \{0,1\}^d} |\mathcal{D}(x) - \mathcal{D}(x^{\sim i})| \\
&= \sum_x \left| \sum_{\ell \in T} p_\ell \left(\mathbb{1}\{x \in \ell\} - \mathbb{1}\{x^{\sim i} \in \ell\}\right)\right|.
\end{aligned}
$$

If $i$ is not queried by any internal node of $T$, then $x, x^{\sim i}$ always belongs to the same leaf and this value is 0. Otherwise, we can trivially bound this value by 2 using the triangle inequality. This discussion leads to the following observation.

**Fact E.3.** *If $f_{\mathcal{D}}$ is computable by a depth-$\ell$ decision tree, then its total influence is at most*

$$\mathrm{Infl}(f_{\mathcal{D}}) \leq 2\ell.$$

For monotone distributions over the Boolean hypercube, influences have a convenient form:

**Proposition E.4** (Lemma 6.2 in Blanc et al. [2023]). *If the distribution $\mathcal{D}$ over $\{0,1\}^d$ is monotone, then for any $i \in [d]$*

$$\mathrm{Infl}_i(f_{\mathcal{D}}) = \mathbb{E}_{\mathcal{D}}[x_i].$$

**Definition E.5** (Restrictions). *Given a sequence of (coordinate, value) pairs $\pi = \{(i_1, b_1), ..., (i_k, b_k)\}$, we use $x_\pi$ to represent $x$ with the coordinates in $\pi$ overwritten/inserted with their respective values. For a function $f : \{0,1\}^d \to \mathbb{R}$, we let $f_\pi$ be the function that maps $x$ to $f(x_\pi)$.*

**Definition E.6** (Everywhere $\tau$-influential; Definition 4 in Blanc et al. [2022b]). *For any function $f : \{0,1\}^d \to \mathbb{R}$, influence threshold $\tau > 0$ and decision tree $T : \{0,1\}^d \to \mathbb{R}$, we say that $T$ is everywhere $\tau$-influential with respect to some $f$ if for every internal node $v$ of $T$, we have*

$$\mathrm{Infl}_{i(v)}(f_v) \geq \tau.$$

*Here $i(v)$ denotes the variable queried at $v$ and $f_v$ denotes the restriction of $f$ by the root-to-$v$ path in $T$, i.e. the variables in the path are fixed to the corresponding internal vertex values.*

### E.2 Replicable Proper Learner for Decision Tree distributions

Our algorithm that achieves the bounds of Theorem 3.2 proceeds in a two-stage manner: we first replicably learn the decision tree structure of $\mathcal{D}$ and then use the replicable uniform-distribution learner to learn $f$ restricted to each of the leaves of the tree. To carry out the first stage, we give an algorithm that replicably learns the optimal decision tree decomposition of a distribution $\mathcal{D}$:

**Theorem E.7** (Replicably Learning DT distributions). *Fix $\alpha, \rho \in (0,1)$ and $\beta \in (0, \rho/3)$. Let $\mathcal{D}$ be a distribution over $\{0,1\}^d$ that is representable by a depth-$\ell$ decision tree. There is an algorithm that*

- *(a) is $\rho$-replicable with respect to $\mathcal{D}$,*
- *(b) returns a depth-$\ell$ tree representing a distribution $\mathcal{D}'$ such that $\mathrm{TV}(\mathcal{D}, \mathcal{D}') \leq \alpha$ with probability at least $1 - \beta$ over the draw of samples,*
- *(c) and has running time and sample complexity*

$$N = \mathrm{poly}(d, 1/\alpha, 1/\rho, \log(1/\beta)) \cdot (2\ell/\alpha)^{O(\ell)}.$$

*For monotone distributions, the algorithm only uses random samples from $\mathcal{D}$, and for general distributions, it uses subcube conditional samples (cf. Definition E.12).*

*Proof (Theorem E.7).* The proof of this result follows directly by applying either Lemma E.11 (for monotone distributions) or Lemma E.13 (for general distributions with subcube conditional sample access) to Theorem E.8, which are provided right after. $\square$

Essentially, making the decision tree learning routine replicable boils down to estimating influences on $\mathcal{D}$ in a replicable manner. This is the content of the upcoming results.

**Theorem E.8** (Replicable Influence $\Rightarrow$ Replicable Decision Tree)**.** *Let $\mathcal{D}$ be a distribution that is representable by a depth-$\ell$ decision tree. For any $\rho' \in (0,1)$ assume access to a $\rho'$-replicable algorithm* rInflEst *for estimating influences of distributions over $\{0,1\}^d$ using $\mathrm{poly}(d, 1/\alpha, 1/\rho', \log(1/\beta))$ samples and runtime. Then, for any $\rho \in (0,1)$, there is an algorithm* rBuildDT *(cf. Algorithm E.1) that*

- *(a) is $\rho$-replicable with respect to $\mathcal{D}$,*
- *(b) returns a depth-$\ell$ tree representing a distribution $\mathcal{D}'$ such that $\mathrm{TV}(\mathcal{D}, \mathcal{D}') \leq \alpha$, with probability at least $1 - \beta$ over the draw of samples,*
- *(c) and has running time and sample complexity*

$$
M = \mathrm{poly}\left(d, \frac{1}{\alpha}, \frac{d(2\ell/\alpha)^{O(\ell)}}{\rho}, \log \frac{d(2\ell/\alpha)^{O(\ell)}}{\beta}\right) \cdot d \cdot (2\ell/\alpha)^{O(\ell)}
$$
$$
+ O\left(\frac{(2\ell/\alpha)^{O(\ell)}}{\alpha^2 \rho^2} \log \frac{(2\ell/\alpha)^{O(\ell)}}{\beta}\right) \cdot (2\ell/\alpha)^{O(\ell)}
$$
$$
= \mathrm{poly}(d, 1/\alpha, 1/\rho, \log(1/\beta)) \cdot (2\ell/\alpha)^{O(\ell)}.
$$

---

**Algorithm E.1** Replicably Building Decision Tree for $\mathcal{D}$

---

1: rBuildDT:
2: **Input:** Access to $\mathcal{D}$ and $f_{\mathcal{D}}$ as in Definition E.1, restriction $\pi$ (cf. Definition E.5), access to rInflEst, depth $\ell$, influence threshold $\tau$, accuracy $\alpha$, confidence $\beta$, replicability $\rho$.
3: **Output:** A decision tree $T$ that minimizes $\mathbb{E}_{t \in T}[\mathrm{Infl}((f_{\mathcal{D}})_t)]$, where $t \in T$ is the collection of leaves (each corresponding to some restriction) of $T$ among all depth-$\ell$, everywhere $\tau$-influential (cf. Definition E.6) trees.
4: **for** $i \in [d]$ **do**
5:    rInflEst$[(f_{\mathcal{D}})_\pi, i] \leftarrow$ rInflEst $\left((f_{\mathcal{D}})_\pi, i, \min(\tau/4, \alpha/2d), \frac{\beta}{2d(2\ell/\alpha)^{O(\ell)}}, \frac{\rho}{2d(2\ell/\alpha)^{O(\ell)}}\right)$
6: **end for**
7: Let $S \subseteq [d]$ be the set of variables $i$ so that rInflEst$[(f_{\mathcal{D}})_\pi, i] \geq 3\tau/4$.
8: **if** $S = \emptyset$ or $\ell = 0$ **then**
9:    // replicably estimate $p_\pi := 2^{|\pi|} \Pr_{x \sim \mathcal{D}}[x$ is consistent with $\pi]$     (cf. Theorem B.4)
10:    $\hat{p}_\pi \leftarrow \alpha/2$-accurate, $\rho/[2(2\ell/\alpha)^{O(\ell)}]$-replicable, $\beta/[2(2\ell/\alpha)^{O(\ell)}]$-confident estimate of $p_\pi$
11:    **return** a new leaf node with value $\hat{p}_\pi$
12: **else**
13:    **for** $i \in S$ **do**
14:       construct the tree $T_i$ with
15:       $\mathrm{root}(T_i) \leftarrow x_i$
16:       $\mathrm{leftSubtree}(T_i) \leftarrow$ rBuildDT$(\mathcal{D}, \pi \cup \{x_i = -1\}, \ell - 1, \tau, \alpha, \beta, \rho)$
17:       $\mathrm{rightSubtree}(T_i) \leftarrow$ rBuildDT$(\mathcal{D}, \pi \cup \{x_i = 1\}, \ell - 1, \tau, \alpha, \beta, \rho)$
18:    **end for**
19:    // Among $\{T_i\}_{i \in S}$, return the one that minimizes the estimated total influence
20:    // $\mathbb{E}_{t \in T_i}\left[\sum_{j \in [d]} \text{rInflEst}[(f_{\mathcal{D}})_t, j]\right]$ where $(f_{\mathcal{D}})_t = (f_{\mathcal{D}})_{\pi(t)}$
21:    // for the restriction $\pi(t)$ corresponding to path from the root the leaf $t$
22:    // Let $g(t) := \sum_{j \in [d]} \text{rInflEst}[(f_{\mathcal{D}})_t, j]$
23:    $i^* \leftarrow \arg\min_{i \in S} \mathbb{E}_{t \in T_i}[g(t)] = \sum_{t \in T_i} 2^{-|t|} g(t) = \sum_{x \in \{0,1\}^d} 2^{-d} g(t) \mathbb{1}\{x \in t\}$
24:    **return** $T_{i^*}$
25: **end if**

---

Before proving Theorem E.8, we state Theorem 3 of Blanc et al. [2023], from which the correctness of Algorithm E.1 follows. Let BuiltDT be the algorithm obtained from rBuildDT (cf. Algorithm E.1) with the following adjustments.

(i) Replace the replicable influence estimator rInflEst with any (possibly non-replicable) influence estimator and

(ii) replace the replicable mean estimator of $p_\pi$ with a simple (non-replicable) sample mean.

**Proposition E.9** (Theorem 3 and Claim 5.5 in Blanc et al. [2023]). *Fix $\alpha, \beta \in (0, 1)$. Let $\mathcal{D}$ be a distribution that is representable by a depth-$\ell$ decision tree. Given oracle access to an estimator for influences, the algorithm* BuildDT *with the choice of $\tau = \alpha/8\ell^2$ returns a depth-$\ell$ tree representing a distribution $\mathcal{D}'$ such that $\mathrm{TV}(\mathcal{D}, \mathcal{D}') \leq \alpha$ with probability at least $1 - \beta$. If the oracle terminates in unit time, the running time of the algorithm is $d \cdot (\ell/\alpha)^{O(\ell)}$.*

**Remark E.10.** *A few remarks are in order regarding Algorithm E.1.*

1. *Blanc et al. [2023] identify the task of learning of a decision tree for a distribution $\mathcal{D}$ with learning its* scaled *pmf $f = 2^d \mathcal{D}$. In order to remain consistent with their convention, the values at the tree leaves produced by our replicable decision tree algorithm (cf. Algorithm E.1) also correspond to the scaled pmf. Note that we do not directly use the values at the leaves and hence this convention is immaterial to our application.*

2. *Estimating the influences to an accuracy of $\tau/4$ ensures that any variables in $S$ has influence at least $\tau/2$ and every variable with influence at least $\tau$ is captured in $S$. Estimating the influences to an accuracy of $\alpha/2d$ ensures that the estimated total influences are accurate up to error at most $\alpha$. Note that the computation for $\mathbb{E}_{t \in T_i}[g(t)]$ (cf. Definition D.6) does not incur additional error since the weights of the sum are appropriately chosen.*

3. *$p_\pi$ is the mean of a $[0, 2^{|\pi|}]$-bounded random variable. By an Hoeffding bound and Theorem B.4, consuming $\mathrm{poly}(2^\ell, 1/\alpha, 1/\rho', \log(1/\beta'))$ samples from $\mathcal{D}$ suffices to estimate this quantity up to $\alpha/[2 \cdot 2^{|\pi|}]$-accuracy, $\rho'$-replicability, and $\beta'$-confidence.*

4. *Taking into account the estimation error of influences, each variable in $S$ has influence at least $\tau/2$. For any depth-$\ell$ decision tree, the sum of all variable influences is at most $2\ell$ (cf. Fact E.3). For any restriction $\pi$, $(f_\mathcal{D})_\pi$ is a depth-$\ell$ decision tree and thus has at most $4\ell/\tau$ variables of influence at least $\tau/2$. For the choice of $\tau = \alpha/8\ell^2$, there can be at most $(4\ell/\tau)^\ell = (2\ell/\alpha)^{O(\ell)}$ recursive calls within Algorithm E.1.*

   *Each recursive call estimates $d$ influences and at most 1 mean of a bounded random variable. All in all, we need to replicably estimate $d \cdot (2\ell/\alpha)^{O(\ell)}$ variable influences and at most $(2\ell/\alpha)^{O(\ell)}$ bounded means.*

We are now ready to prove Theorem E.8.

*Proof (Theorem E.8).* By assumption, for any $\rho' \in (0, 1)$, rInflEst is $\rho'$-replicable and returns $\alpha$-accurate estimates for influences with confidence $\beta$ using $\mathrm{poly}(d, 1/\alpha, \log(1/\beta), 1/\rho')$ samples and runtime.

The replicability of Algorithm E.1 follows directly from the replicability of the influence estimators and the replicability of the standard rounding scheme (cf. Theorem B.4). By Remark E.10, it suffices to call the replicable influence estimator and the replicable mean estimator for $p_\pi$ with replicability parameter $\rho' = \rho/[2d(2\ell/\alpha)^{O(\ell)}]$ for the union bound and similarly for the confidence parameter. The sample complexity follows from our remark above.

Let us now condition on the event that each call of rInflEst and estimation of expected total influence is successful. Conditioned on this event, the correctness of the algorithm rBuildDT directly follows from Proposition E.9. In particular, if we condition on the event that each call to rInflEst and each replicable estimate of $p_\pi$ is successful, our algorithm reduces to the BuildDT algorithm of Blanc et al. [2023].

$\square$

### E.3 Replicable Influence Estimator for Monotone Marginals

In this section, we provide the main subroutine of our replicable DT distribution learning algorithm. We begin with a replicable influence estimator for monotone distributions.

**Lemma E.11** (Replicable Influence Estimator for Monotone Marginals). *Fix $i \in [d]$. For any $\alpha, \beta, \rho \in (0, 1)^3$, there is an efficient $\rho$-replicable algorithm* rInflEst$(\mathcal{D}, i, \alpha, \beta, \rho)$ *such that given*

*an unknown monotone distribution $\mathcal{D}$ over $\{0,1\}^d$, computes an estimate of $\mathrm{Infl}_i(f_\mathcal{D})$ up to accuracy $\alpha$ with probability at least $1 - \beta$ using $O(\alpha^{-2}\rho^{-2}\log(1/\beta))$ time and random samples from $\mathcal{D}$.*

*Proof (Lemma E.11).* By Proposition E.4, since $\mathcal{D}$ is monotone, we know that $\mathrm{Infl}_i(f_\mathcal{D}) = \mathbb{E}_\mathcal{D}[x_i]$. Thus we can replicably estimate this expectation with $O(\alpha^{-2}\rho^{-2}\log(1/\beta))$ samples and sample-polynomial runtime through replicable query rounding (cf. Theorem B.4). $\qquad\square$

### E.4   Replicable Influence Estimator for Arbitrary Distributions

We further design a replicable influence estimator for arbitrary distributions $\mathcal{D}$ over $\{0,1\}^d$ using subcube conditional sampling.

**Definition E.12.** *A (subcube) conditional sampling oracle for distribution $\mathcal{D}$ over $\{0,1\}^d$ receives as input a subset (subcube) $S \subseteq \{0,1\}^d$ and generates a sample from the conditional distribution*

$$\mathcal{D}_S(x) := \mathcal{D}(x \mid S) = \frac{\mathcal{D}(x)\mathbb{1}\{x \in S\}}{\mathcal{D}(S)}.$$

This conditional distribution is essentially the truncated distribution $\mathcal{D}_S$ with truncation set $S$. The subcube conditioning oracle is well studied by prior work in distribution testing and learning [Canonne et al., 2015, 2021, Fotakis et al., 2022, 2020, Gouleakis et al., 2017].

---

**Algorithm E.2** Replicable Influence Estimation for Arbitrary Distributions via Conditional Sampling

---

1: `rInflEst`:
2: **Input:** Distribution $\mathcal{D}$ over $\{0,1\}^d$ and $f_\mathcal{D}$ as in Definition E.1, coordinate $i \in [d]$, access to conditional sampling oracle for $\mathcal{D}$, desired accuracy $\alpha$, desired confidence $\beta$, desired replicability $\rho$.
3: **Output:** A estimate $v$ of $\mathrm{Infl}_i(f_\mathcal{D})$ such that $|v - \mathrm{Infl}_i(f_\mathcal{D})| \le \alpha$ with probability $1 - \beta$.
4: $n \leftarrow O(\alpha^{-2}\rho^{-2}\log(1/\beta))$
5: **for** $j \leftarrow 1,...,n$ **do**
6:     Draw a fresh random sample $x^{(j)} \sim \mathcal{D}$
7:     $S^{(j)} \leftarrow \{x^{(j)}\} \cup \{(x^{(j)})^{\sim i}\}$, where $(x^{(j)})^{\sim i}$ is $x^{(j)}$ with its $i$-th coordinate flipped
8:     Draw $O(\alpha^{-2})$ independent samples from the conditional on $S^{(j)}$ distribution $\mathcal{D}(\cdot|S^{(j)})$
9:     Let $p^{(j)} \in [0,1]$ be the fraction that we observe $x^{(j)}$
10:     $q^{(j)} \leftarrow |p^{(j)} - (1-p^{(j)})|$
11: **end for**
12: $q \leftarrow \frac{1}{n}\sum_{j\in[n]} q^{(j)}$
13: **return** $v \leftarrow$ `rRound`$(q, O(\alpha\rho), \rho)$     (see Theorem B.4 with $T = 1$)

---

An adaptation of the proof of Proposition 6.5 in Blanc et al. [2023] combined with the replicability properties of Theorem B.4 gives the following result, which we provide for completeness but do not use in our applications. The algorithm is presented Algorithm E.2.

**Lemma E.13** (Replicable Influence Estimator for Arbitrary Marginals)**.** *Let $\mathcal{D}$ be a distribution over $\{0,1\}^d$, $f_\mathcal{D}$ as in Definition E.1, and, assume sample access to a subcube conditional oracle as in Definition E.12. For any $\alpha, \beta, \rho \in (0,1)$ and $i \in [d]$, Algorithm E.2 is $\rho$-replicable and, given $O(\alpha^{-4}\rho^{-2}\log(1/\beta))$ subcube conditional samples from $\mathcal{D}$, it computes in sample-polynomial time an estimate $v$ that satisfies $|v - \mathrm{Infl}_i(f_\mathcal{D})| \le \alpha$ with probability at least $1 - \beta$.*

As a proof sketch, let us call the above algorithm `ReplInfEst`$(\mathcal{D}, i, \alpha)$. The correctness of the algorithm follows from the observation that for any distribution $\mathcal{D}$, coordinate $i$ and $\alpha \in (0,1)$, for any execution $j \in [n]$, it holds that $|\mathbb{E}q^{(j)} - \mathrm{Infl}_i(f_\mathcal{D})| \le \alpha$. Hence to obtain a high probability estimator, it suffices to obtain $O(\alpha^{-2}\log(1/\beta))$ copies of $q^{(j)}$ and take the average $q$. Finally, to make the algorithm replicable, it suffices to estimate the average $q$ replicably. This can be accomplished at an extra cost of order $1/\rho^2$ using the standard rounding routines (cf. Theorem B.4). Since each iteration requires $1/\alpha^2$ subcube samples, the sample complexity of our algorithm follows.

### E.5   Useful Subroutines & Results

In this section, we provide a set of useful results that we will use in our proofs.

**Proposition E.14** (Corollary D.21 in Esfandiari et al. [2023b])**.** *Let $\alpha, \rho \in (0,1)$ and $\beta \in (0, \rho/3)$. There is a $\rho$-replicable algorithm `rFiniteDistrEst`$(\mathcal{D}, \alpha, \beta, \rho)$ that outputs parameter estimates $\bar{p}$ for a finite distribution $\mathcal{D}$ of support size $N$ such that*

*(a)* $|\bar{p}^{(i)} - p^{(i)}| \le \alpha$ *for every* $i \in [N]$ *with probability at least* $1 - \beta$.

*(b)* $\bar{p}^{(i)} \ge 0$ *for all* $i \in [N]$.

*(c)* $\sum_i \bar{p}^{(i)} = 1$.

*Moreover, the algorithm has sample complexity*

$$m = O\left(\frac{\ln{1/\beta} + N}{\alpha^2 (\rho - \beta)^2}\right) = O\left(\frac{N}{\alpha^2 \rho^2} \log \frac{1}{\beta}\right)$$

*and* $\mathrm{poly}(m)$ *time complexity.*

**Proposition E.15** ([Janson, 2018]). *Let* $Y_i$ *be a geometric variable with success rate* $q$. *Then* $Y := \sum_{i=1}^{m} Y_i$ *is the number of draws until we obtain* $m$ *successes. Then*

$$\mathbb{P}\left\{Y \ge \lambda \frac{m}{q}\right\} \le \exp(1 - \lambda).$$

In other words, it suffices to perform $O(m/q \log(1/\beta))$ Poisson trials before succeeding $m$ times with probability at least $1 - \beta$.

**Proposition E.16** (Replicable Boosting of Success Probability). *Let* $\alpha, \rho \in (0, 1)$, $\beta \in (0, \rho/3)$, *and* $\Delta \ge 0$. *Suppose* $\mathcal{A}$ *is a* $\rho'$-*replicable* $(\alpha + \Delta, 1/2)$-*PAC learner for the concept class* $\mathscr{C}$ *under a fixed distribution* $\mathcal{D}$ *over* $\{0,1\}^d$ *that uses* $m(\alpha, \rho') = \mathrm{poly}(d, 1/\alpha, 1/\rho')$ *samples for any* $\rho' \in (0, 1)$. *There is a* $\rho$-*replicable* $(2\alpha + \Delta, \beta)$-*PAC learner* $\mathtt{rBoost}_{\mathcal{A}}(\mathcal{D}, \alpha, \beta, \rho)$ *for* $\mathscr{C}$ *under* $\mathcal{D}$ *for any* $\alpha, \beta, \rho \in (0, 1)$ *that has sample complexity*

$$m\left(d, \frac{\rho}{2 \log(1/\beta)}\right) \cdot O\left(\log \frac{1}{\beta}\right) + O\left(\frac{\log^2(1/\beta)}{\alpha^2 \rho^2} \log \frac{\log(1/\beta)}{\beta}\right) = \mathrm{poly}\left(d, 1/\alpha, 1/\rho, \log(1/\beta)\right).$$

*and sample-polynomial running time.*

*Proof.* Let $\mathtt{rBoost}_{\mathcal{A}}$ be the algorithm that runs $\mathcal{A}$ for $n = O(\log 1/\beta)$ times (each with $m = m(1/\alpha, \rho/2n)$ samples), replicably estimates the population error of each hypothesis, and outputs the hypothesis which has the lowest estimated error.

Running $\mathcal{A}$ for $n = O(\log 1/\beta)$ times with $m$ samples per run guarantees that each execution is $\rho/2n$-replicable so the first step is $\rho/2$-replicable. Moreover, at least one of the hypotheses we output is $(\alpha + \Delta)$-close to the optimal hypothesis with probability at least $1 - \beta/2$.

By an Hoeffding bound on the empirical error of a hypothesis, we can estimate the population error of a hypothesis to an accuracy of $\alpha\rho/4 \cdot 2n$ and confidence $\beta/2n$ using $O(n^2 \alpha^{-2} \rho^{-2} \log(n/\beta))$ samples. By Theorem B.4, we can make this estimate $\rho/2n$-replicable at the cost of reducing the accuracy to $\alpha$ while maintaining the confidence parameter $\beta/2n$.

It follows that the output hypothesis is $\rho$-replicable and with probability at least $1 - \beta$, its population error is at most $2\alpha + \Delta$. $\qquad\square$

### E.6 Proof of Theorem 3.2

In this section, we show how to lift replicable algorithms that learn over the uniform distribution to replicable algorithms that learn with respect to arbitrary distributions, where the sample complexity and running time depend on the decision tree complexity of the target distribution. Let $\mathcal{D}(\cdot \mid T)$ denote the conditional distribution on leaves, i.e., $\mathcal{D}(t \mid T) := \sum_{x \in t} \mathcal{D}(x)$.

Now, our learning algorithm is designed for an exact tree distribution. However, we can only estimate the tree representation of the actual distribution, i.e., the conditional distribution at a leaf subcube is uniform in the tree representation but not necessarily uniform in the input distribution. To show that our learning algorithm can accommodate slight errors in the tree distribution estimation, we introduce the following definition.

**Definition E.17** (Robust Learning). *For any concept class* $\mathscr{C}$ *and algorithm* $\mathcal{A}$, *we say that* $\mathcal{A}$ $(\alpha, \beta, c)$-*robustly learns* $\mathscr{C}$ *using* $m$ *samples under the uniform distribution if for any* $\eta > 0$ *and class*

$$\mathscr{D}_{\eta} = \{\text{distribution } \mathcal{D} \text{ over } \{0,1\}^d : \mathrm{TV}(\mathcal{U}, \mathcal{D}) \le \eta\},$$

*it holds that* $\mathcal{A}(\alpha + c\eta, \beta)$-*learns* $\mathscr{C}$ *using* $m$ *samples with respect to the distributions in* $\mathscr{D}_{\eta}$.

**Proposition E.18** (Proposition 7.2 from [Blanc et al., 2022b]). *For any concept class* $\mathscr{C}$ *and algorithm* $\mathcal{A}$, *if* $\mathcal{A}$ $(\alpha, \beta)$-*PAC learns* $\mathscr{C}$ *using* $m$ *samples under the uniform distribution, then the exact same algorithm* $\mathcal{A}$ *also* $(\alpha, \beta + 1/3, 3m)$-*robustly learns* $\mathscr{C}$ *using* $m$ *samples.*

---

**Algorithm E.3** Replicable Lifting

---

1: rLift
2: **Input:** decision tree $T$ (from rBuildDT), algorithm $\mathcal{A}$ for $\rho'$-replicable $(\alpha', \beta')$-learning under the uniform distribution using $m(\alpha', \rho', \beta')$ samples, distribution $\mathcal{D}$, accuracy $\alpha$, confidence $\beta$, replicability $\rho$.
3: **Output:** A hypothesis $h : \{0,1\}^d \to \{0,1\}$
4:
5: $M_1 \leftarrow \text{poly}(2^\ell, 1/\alpha, 1/\rho, \log(1/\beta))$
6: $M_2 \leftarrow m(\alpha/6, \rho/2^\ell, 1/6) \cdot \text{poly}(d, 2^\ell, 1/\alpha, 1/\rho, \log(1/\beta))$
7: // Estimate leaf distribution (cf. Proposition E.14)
8: $\hat{\mathcal{D}}(\cdot \mid T) \leftarrow \texttt{rFiniteDistrEst}(\mathcal{D}(\cdot \mid T), \alpha/12 \cdot 2^\ell, \beta/2, \rho/3)$ using $M_1$ samples
9: Draw a dataset $S \sim \mathcal{D}^{M_2}$ of size $M_2$.
10: **for** each leaf $t \in T$ **do**
11:     **if** $\hat{\mathcal{D}}(t \mid T) \geq \alpha/4 \cdot 2^\ell$ **then**
12:         Let $S_t$ be the subset of samples in $S$ that reach $t$.
13:         Create a set $S_t'$ consisting of points in $S_t$ but where all coordinates queried on the root-to-leaf path for $t$ are re-randomized independently, making the marginal distribution uniform.
14:         For $\rho' \in (0,1)$, choose its parameters so that $\mathcal{A} = \mathcal{A}(\alpha/6, \rho', 1/6)$ is $\rho'$-replicable and $(\alpha/6, 1/6 + 1/3, c')$-robustly learns $\mathscr{C}$.
15:         Get $h_t$ by calling $\texttt{rBoost}_\mathcal{A}(S_t', \alpha/6, \rho/3 \cdot 2^\ell, \beta/2 \cdot 2^\ell)$ (cf. Proposition E.16).
16:     **else**
17:         Get $h_t$ that outputs a random guess according to shared randomness.
18:     **end if**
19: **end for**
20: Return the hypothesis $h$ such that given input $x$, finds the leaf $t \in T$ that $x$ follows and outputs $h_t(x)$.

---

We are now ready to state the main result of this section, from which the proof of Theorem 3.2 closely follows.

**Theorem E.19.** *Consider a concept class $\mathscr{C}$ of functions $f : \{0,1\}^d \to \{0,1\}$ closed under restrictions. Fix $\alpha, \beta, c, \rho > 0$ and $m, \ell \in \mathbb{N}$. Suppose we are provided black-box access to an algorithm $\mathcal{A}$ that*

    *(i) is $\rho'$-replicable with respect to the uniform distribution for any $\rho' \in (0,1)$,*
    *(ii) $(\alpha', \beta' + 1/3, c)$-robustly learns $\mathscr{C}$ for any $\alpha', \beta' \in (0,1)$, and*
    *(iii) consumes $m(\alpha', \rho', \beta') = \text{poly}(d, 1/\alpha', 1/\rho', \log(1/\beta'))$ samples and computation time under the uniform distribution.*

*Let $M_1 = \text{poly}(2^\ell, 1/\alpha, 1/\rho, \log(1/\beta))$, $M_2 = m(\alpha/6, \rho/2^\ell, 1/6) \cdot \text{poly}(d, 2^\ell, 1/\alpha, 1/\rho, \log(1/\beta))$, and*

$$M := M_1 + M_2 \leq \text{poly}(d, 2^\ell, 1/\alpha, 1/\rho, \log(1/\beta)).$$

*For any function $f^\star \in \mathscr{C}$, distribution $\mathcal{D}$ over $\{0,1\}^d$, depth-$\ell$ decision tree $T$ computing the pmf of a distribution $\mathcal{D}_T$ where $\text{TV}(\mathcal{D}, \mathcal{D}_T) \leq \alpha/6c$, the following holds.*
*The algorithm $\texttt{rLift}(T, \mathcal{A}, \mathcal{D}, \alpha, \beta, \rho)$ (cf. Algorithm E.3) is $\rho$-replicable, has $O(M)$ sample complexity, terminates in $\text{poly}(M)$ time, and its output is $\alpha$-close to $f^\star$ with respect to $\mathcal{D}$ with probability at least $1 - \beta$.*

Before we prove Theorem E.19, we state two useful lemmas.

**Lemma E.20.** *Fix $\alpha, \rho, \rho', \beta \in (0,1)$, $\Delta \geq 0$, and $\ell \in \mathbb{N}$. Let $m_{\texttt{rBoost}} = \text{poly}(d, 2^\ell, 1/\alpha, 1/\rho, \log(1/\beta))$ be the number of samples that $\texttt{rBoost}$ (cf. Proposition E.16) requires to boost a $\rho'$-replicable $(\alpha/6 + \Delta, 1/2)$-correct learner (that uses $\text{poly}(d, 1/\alpha, 1/\rho')$ samples and running time) to a $\rho/3 \cdot 2^\ell$-replicable learner with accuracy $\alpha/6 + \Delta + \alpha/6 = \alpha/3 + \Delta$ and confidence $\beta/2 \cdot 2^\ell$. Condition on the success of $\texttt{rFiniteDistrEst}$ in Algorithm E.3. Then with probability at least $1 - \beta/2$, for every $t \in T$ with $\hat{\mathcal{D}}(t \mid T) \geq \alpha/4 \cdot 2^\ell$, we observe at least $m_{\texttt{rBoost}}$ samples reaching $t$ from $S$.*

*Proof.* Condition on the success of the call to $\texttt{rFiniteDistrEst}$. Then for every $t \in T$ with $\hat{\mathcal{D}}(t \mid T) \geq \alpha/4 \cdot 2^\ell$, we have $\mathcal{D}(t \mid T) \geq \alpha/6 \cdot 2^\ell$. The probability of not observing such a $t$ in a dataset

of size $n$ is at most $(1 - \alpha/6 \cdot 2^\ell)^n \leq \exp(-n\alpha/6 \cdot 2^\ell)$. By a union bound over the at most $6 \cdot 2^\ell/\alpha$ such $t$, the probability of failing to collect each such $t$ is at most

$$\frac{6 \cdot 2^\ell}{\alpha} \exp(-n\alpha/6 \cdot 2^\ell).$$

To drive this below the constant $1/2$, it suffices to take $n = O(\ell \cdot 2^\ell/\alpha \log 1/\alpha)$. By Proposition E.15, it suffices to take a sample of size

$$O(n m_{\texttt{rBoost}} \log 1/\beta) = O\left( \frac{m_{\texttt{rBoost}} \ell \cdot 2^\ell}{\alpha} \left( \log \frac{1}{\alpha} \right) \left( \log \frac{1}{\beta} \right) \right) \leq M_2$$

to collect $m$ copies of each such $t$ with probability at least $1 - \beta/2$. $\qquad\square$

**Lemma E.21** (Lemma B.4 from [Blanc et al., 2022a], Fact 7.4 from [Blanc et al., 2023]). *For any distribution $\mathcal{D}$ and decision tree $T$ computing the pmf of another distribution $\mathcal{D}_T$,*

$$\sum_{t \in T} \Pr_{x \sim \mathcal{D}} [x \text{ reaches } t] \cdot \mathrm{TV}(\mathcal{D}_t, (\mathcal{D}_T)_t) \leq 2\mathrm{TV}(\mathcal{D}, \mathcal{D}_T),$$

*where $(\mathcal{D}_T)_t$ (resp. $\mathcal{D}_t$) is the conditional of $\mathcal{D}_T$ (resp. $\mathcal{D}$) on the subcube induced by the leaf $t$.*

Note that in the statement of the lemma above, $\Pr_{x \sim \mathcal{D}} [x \text{ reaches } t] = \mathcal{D}(t \mid T)$ and $(\mathcal{D}_T)_t$ is the uniform distribution on the subcube represented by the leaf $t$. We are finally ready to prove Theorem E.19.

*Proof (Theorem E.19).* We first note that the sample complexity of the call to the routine $\texttt{rFiniteDistrEst}$ is $M_1 = \mathrm{poly}(2^\ell, 1/\alpha, 1/\rho, \log(1/\beta))$ (cf. Proposition E.14). Throughout this proof, we condition on the $1 - \beta/2$ probability of $\texttt{rFiniteDistrEst}$ succeeding.
Let $h$ be the output of $\texttt{rLift}(T, \mathcal{A}, S)$. We now argue separately about the accuracy and replicability of the algorithm.

**Accuracy.** The accuracy of the learner $h = \texttt{rLift}(T, \mathcal{A}, S)$ is analyzed as follows. For any leaf $t$ of the tree $T$, let $h_t$ be the associated predictor. We have that

$$\Pr_{x \sim \mathcal{D}}[h(x) \neq f^\star(x)] = \sum_{t \in T} \Pr_{x \sim \mathcal{D}} [x \text{ reaches } t] \Pr_{x \sim \mathcal{D}_t} [h_t(x) \neq f^\star(x)].$$

Each hypothesis $h_t$ is obtained either by running $\texttt{rBoost}_{\mathcal{A}}$ (cf. Proposition E.16) on the sample $S'_t$ given that $\hat{\mathcal{D}}(t \mid T) \geq \alpha/4 \cdot 2^\ell$ or corresponds to a random guess otherwise.
On the other hand, from the choice of estimation error to the call for $\texttt{rFiniteDistrEst}$, each leaf $t$ on which we output the random guess hypothesis must satisfy $\Pr_{x \sim \mathcal{D}}[x \text{ reaches } t] < \alpha/3 \cdot 2^\ell$. The overall error contribution of such leaves is thus at most $\alpha/3$. It follows that total error is at most

$$\Pr_{x \sim \mathcal{D}}[h(x) \neq f^\star(x)] \leq \sum_{t \in T : \hat{\mathcal{D}}(t|T) \geq \alpha/4 \cdot 2^\ell} \Pr_{x \sim \mathcal{D}} [x \text{ reaches } t] \Pr_{x \sim \mathcal{D}_t} [h_t(x) \neq f^\star(x)] + \frac{\alpha}{3}.$$

Fix a $t \in T$ on which we run $\texttt{rBoost}_{\mathcal{A}}$. Recall that $\mathcal{D}_t$ and $(\mathcal{D}_T)_t$ denotes the conditional distribution on the leaf subcube over the coordinates not fixed by $\pi = \pi(t)$ where $\pi$ is the restriction corresponding to $t$. Here the underlying distributions are the input distribution $\mathcal{D}$ and tree distribution $\mathcal{D}_T$. Let $\mathcal{D}'_t, (\mathcal{D}_T)'_t$ denote the distributions over $\{0,1\}^d$ obtained from $\mathcal{D}_t, (\mathcal{D}_T)_t$ by re-randomizing the coordinates from $\pi$. Then $(\mathcal{D}_T)'_t$ is precisely the uniform distribution $\mathcal{U}$ over $\{0,1\}^d$ and $\mathcal{D}'_t$ satisfies

$$
\begin{aligned}
&\mathrm{TV}(\mathcal{D}'_t, \mathcal{U}) \\
&= \mathrm{TV}(\mathcal{D}'_t, (\mathcal{D}_T)'_t) \\
&:= \frac{1}{2} \sum_{x \in \{0,1\}^d} |\mathcal{D}'_t(x) - (\mathcal{D}_T)'_t(x)| \\
&= \frac{1}{2} \sum_{x \in t} 2^{|\pi(t)|} |\mathcal{D}'_t(x) - (\mathcal{D}_T)'_t(x)| \\
&= \frac{1}{2} \sum_{x \in t} |\mathcal{D}_t(x) - (\mathcal{D}_T)_t(x)| \qquad \mathcal{D}'_t(x) = 2^{-|\pi(t)|} \mathcal{D}_t(x), (\mathcal{D}_T)'_t(x) = 2^{-|\pi(t)|} (\mathcal{D}_T)_t(x) \\
&= \mathrm{TV}(\mathcal{D}_t, (\mathcal{D}_T)_t).
\end{aligned}
$$

Based on the re-sampling step of the algorithm, we know that any point in $S'_t$ is an i.i.d. sample from $\mathcal{D}'_t$ and labeled by the function $f^\star_t$, which lies in $\mathscr{C}$ thanks to closedness under restrictions.

By $(\alpha/6, 1/6 + 1/3, c)$-robust learnability, $\mathcal{A}$ in fact $(\alpha/6 + c\mathrm{TV}(\mathcal{D}'_t, \mathcal{U}), 1/2)$-PAC learns $\mathscr{C}$ under the distribution $\mathcal{D}'_t$. We now apply Lemma E.20 with $\Delta = c\mathrm{TV}(\mathcal{D}'_t, \mathcal{U})$: the leaves on which we run $\mathtt{rBoost}_\mathcal{A}$ have at least $m_{\mathtt{rBoost}}$ samples where $m_{\mathtt{rBoost}} = \mathrm{poly}(d, 2^\ell, 1/\alpha, 1/\rho, \log(1/\beta))$ is the number of samples that $\mathtt{rBoost}$ requires to boost a $\rho'$-replicable $(\alpha/6 + c\mathrm{TV}(\mathcal{D}'_t, \mathcal{U}), 1/2)$-correct learner (that uses $\mathrm{poly}(d, 1/\alpha, 1/\rho')$ samples and running time) to a $\rho/3 \cdot 2^\ell$-replicable learner with accuracy $\alpha/6 + c\mathrm{TV}(\mathcal{D}'_t, \mathcal{U}) + \alpha/6 = \alpha/3 + c\mathrm{TV}(\mathcal{D}'_t, \mathcal{U})$ and confidence $\beta/2 \cdot 2^\ell$.

Then by Proposition E.16, with probability at least $1 - \beta/2 \cdot 2^\ell$ over the data $S'_t$, the hypothesis $h_t$ output by $\mathtt{rBoost}_\mathcal{A}$ satisfies

$$\Pr_{x \sim \mathcal{D}_t}[h_t(x) \neq f^\star(x)] \leq \alpha/3 + c\,\mathrm{TV}(\mathcal{D}'_t, \mathcal{U}) = \alpha/3 + c\,\mathrm{TV}(\mathcal{D}_t, (\mathcal{D}_T)_t).$$

All in all, Combined with the expression of total error above, this means that with probability at least $1 - \beta$,

$$\Pr_{x \sim \mathcal{D}}[h(x) \neq f^\star(x)] \leq 2\alpha/3 + c \sum_{t \in T} \Pr_{x \sim \mathcal{D}}[x \text{ reaches } t]\,\mathrm{TV}(\mathcal{D}_t, (\mathcal{D}_T)_t).$$

We finish by applying Lemma E.21, which ensures that the sum over leaves above is upper bounded by $2\mathrm{TV}(\mathcal{D}, \mathcal{D}_T) \leq \alpha/3c$ and so the total misclassification error is at most $\alpha$ with probability at least $1 - \beta$.

**Replicability.**  We assume we are given the same decision tree in both executions. Next, we note that $\mathtt{rFiniteDistrEst}$ is $\rho/3$-replicable and succeeds in two executions with probability at least $1 - 2 \cdot \beta/2 \geq 1 - \rho/3$. Conditional on the events above, it suffices to show that for any leaf of the tree, the algorithm $\mathtt{rBoost}_\mathcal{A}$ replicably outputs a hypothesis. Since there are at most $2^\ell$ leaves and each call of $\mathtt{rBoost}_\mathcal{A}$ has replicability parameter $\rho/3 \cdot 2^\ell$, the entire procedure is $\rho$-replicable. $\qquad\square$

Before moving on to the proof of Theorem 3.2, we state one final lemma.

**Lemma E.22.** *For any concept class $\mathscr{C}$ and algorithm $\mathcal{A}$, if $\mathcal{A}$ $\rho$-replicably $(\alpha, \beta)$-learns $\mathscr{C}$ using $m$ samples under the uniform distribution, then the exact same algorithm $\mathcal{A}$ also $\rho$-replicably $(\alpha, \beta + 1/3, 3m)$-robustly learns $\mathscr{C}$ using $m$ samples under the uniform distribution.*

*Proof (Lemma E.22).*  The proof follows directly from the analysis of Proposition 7.2 in Blanc et al. [2023] (cf. Proposition E.18). The replicability of the algorithm is trivially preserved since the algorithm does not change. $\qquad\square$

We restate the statement of Theorem 3.2 below for convenience before its formal proof.

**Theorem 3.2** (Lifting Replicable Uniform Learners)**.** *Consider a concept class $\mathscr{C}$ of functions $f : \{0,1\}^d \to \{0,1\}$ closed under restrictions. Suppose we are given black-box access to an algorithm such that for any $\alpha', \rho', \beta' \in (0,1)$, given $\mathrm{poly}(d, 1/\alpha', 1/\rho', \log(1/\beta'))$ samples from $\mathcal{U}$,*
  *(i) is $\rho'$-replicable with respect to the uniform distribution $\mathcal{U}$,*
  *(ii) PAC learns $\mathscr{C}$ under the uniform distribution to accuracy $\alpha'$ and confidence $\beta'$, and,*
  *(iii) terminates in time $\mathrm{poly}(d, 1/\alpha', 1/\rho', \log(1/\beta'))$.*
*Let $\alpha, \rho \in (0,1)$ and $\beta \in (0, \rho/3)$. For $m = \mathrm{poly}(d, 1/\alpha, 1/\rho, \log(1/\beta))$ and $M = \mathrm{poly}(d, 1/\alpha, 1/\rho, \log(1/\beta))^{O(\ell)}$, the following cases hold:*

(a) *If $\mathcal{D}$ is a monotone distribution over $\{0,1\}^d$ representable by a depth-$\ell$ decision tree, there is an algorithm that draws $M$ samples from $\mathcal{D}$, is $\rho$-replicable with respect to $\mathcal{D}$, PAC learns $\mathscr{C}$ under $\mathcal{D}$ with accuracy $\alpha$ and confidence $\beta$, and terminates in $\mathrm{poly}(M)$ time.*
(b) *If $\mathcal{D}$ is an arbitrary distribution over $\{0,1\}^d$ representable by a depth-$\ell$ decision tree, there is an algorithm that draws $M$ labeled examples as well as $M$ conditional samples (cf. Definition D.5) from $\mathcal{D}$, is $\rho$-replicable with respect to $\mathcal{D}$, PAC learns $\mathscr{C}$ with respect to $\mathcal{D}$ with accuracy $\alpha$ and confidence $\beta$, and terminates in $\mathrm{poly}(M)$ time.*

*Proof (Theorem 3.2).*  Using Theorem E.19, we can prove Theorem 3.2 as follows. First, we use Theorem E.7 to replicably learn the input distribution to total variation distance $\alpha/6 \cdot 3m$ with a decision tree. Next, Lemma E.22 asserts that $\mathcal{A}$ is $\rho$-replicable and $(\alpha, \beta + 1/3, 3m)$-robustly learns $\mathscr{C}$ under the uniform distribution. The result follows by applying Theorem E.19. $\qquad\square$

# F  Efficient Replicability and Private Learning

## F.1  Replicably Learning Finite Classes

We first state a useful result which we later leverage as a subroutine appears in Bun et al. [2023].

**Theorem F.1** (Replicable Learner for Finite Classes; Theorem 5.13 in [Bun et al., 2023]). *Let* $\alpha, \rho, \beta \in (0,1)^3$. *Then, any finite concept class* $\mathscr{C}$ *is replicably learnable in the agnostic setting with*

$$m(\alpha, \rho, \beta) = O\left(\frac{\log^2 |\mathscr{C}| + \log\frac{1}{\rho\beta}}{\alpha^2\rho^2}\log^3\frac{1}{\rho}\right),$$

*many samples. Moreover, the running time of the algorithm is polynomial in the number of samples and the cardinality of* $\mathscr{C}$.

## F.2  A Transformation from Pure DP to Replicability

Our approach borrows some high level ideas from Gonen et al. [2019] who showed a transformation from a pure DP PAC learner to an online learner. In a nutshell, our algorithm works as follows:

1) First, we create a "dummy" input dataset $\bar{S}$ which has some constant size[13] and we run the algorithm for $\widetilde{\Theta}\left(\frac{\log^3(1/\beta)}{\rho^2}\right)$ times on $\bar{S}$, resampling its internal randomness every time.
2) With probability $1 - \beta$, one of the hypotheses will have error at most $3/8$ [Beimel et al., 2013, Gonen et al., 2019].
3) We run the replicable agnostic learner for finite classes from Bun et al. [2023].
4) We boost the weak replicable learner using Impagliazzo et al. [2022], Kalavasis et al. [2023].

We begin by presenting an adaptation of a result that appeared in Beimel et al. [2013], Gonen et al. [2019]. Essentially, it states that when a class is learnable by a pure DP learner, one can fix an arbitrary input $\bar{S}$ that has constant size, execute the learner on this dataset a constant number of times by resampling its *internal* randomness, and then with some constant probability, e.g., $15/16$, the output will contain one hypothesis whose error is bounded away from $1/2$ by some constant, e.g., $1/4$. This result might seem counter-intuitive at first glance, but learnability by a pure DP learner is a very strong property for a class and the result crucially depends on this property. Below, we present an adaptation of this result which states that the weak learner can be $\rho$-replicable and has a probability of success of $1 - \delta$. An important element of our derivation is a result from Bun et al. [2023] about the sample and computational complexity of agnostically learning a concept class using a replicable algorithm (cf. Theorem F.1).

We are now ready to state a key lemma for our transformation.

**Lemma F.2** (From Pure DP Learner to Replicable Weak Learner). *Let* $\mathcal{X}$ *be some input domain,* $\mathcal{Y} = \{0,1\}$, *and* $\mathcal{D}_{XY}$ *be a distribution on* $\mathcal{X} \times \mathcal{Y}$ *that is realizable with respect to some concept class* $\mathscr{C}$. *Let* $\mathcal{A}$ *be a pure DP learner such that for any* $\alpha, \varepsilon, \beta \in (0,1)^3$, $\mathcal{A}$ *needs* $m(\alpha, \varepsilon, \beta, \mathscr{C})$ *i.i.d. samples from* $\mathcal{D}_{XY}$ *to output with probability at least* $1 - \beta$ *a hypothesis with error at most* $\alpha$ *in an* $\varepsilon$-*DP way. Let us set* $m_0 = m(1/4, 1/10, 1/2, \mathscr{C})$.
*Then, for any* $\rho, \beta' \in (0,1)^2$ *there is a* $\rho$-*replicable learner* $\mathcal{A}'$ *that outputs a hypothesis with error at most* $3/8$ *with probability at least* $1 - \beta'$ *and requires* $\widetilde{O}\left(\text{poly}(m_0)\frac{\log^3(1/\beta')}{\rho^2}\right)$ *i.i.d. samples from* $\mathcal{D}_{XY}$ *and* $O(\exp(m_0)\log(1/\beta'))$ *oracle calls to* $\mathcal{A}$ *(and hence runtime).*

*Proof (Lemma F.2).* We argue about the correctness and replicability of the algorithm separately.

**Correctness.**  Since $\mathcal{D}_{XY}$ is realizable, there exists some $c^\star \in \mathscr{C}$ such that $\text{loss}_{\mathcal{D}_{XY}}(c^\star) = 0$. Let

$$\text{range}(\mathcal{A}) = \{c : \mathcal{X} \to \{0,1\} : \exists S \in \cup_{n\in\mathbb{N}}(\mathcal{X} \times \{0,1\})^n, \exists r \in \cup_{n\in\mathbb{N}}\{0,1\}^n, \text{ s.t. } \mathcal{A}(S; r) = c\}.$$

Let also $m_0 = m(1/4, 1/10, 1/2, \mathscr{C})$ be the number of samples $\mathcal{A}$ needs to achieve $\alpha = 1/4, \varepsilon = 1/10, \beta = 1/2$ for the class $\mathscr{C}$. Notice that $m_0$ is a constant with respect to $\alpha, \beta, \rho, \epsilon$. Let also

$$\mathscr{C}(\mathcal{D}_{XY}) = \{c \in \text{range}(\mathcal{A}) : \text{loss}_{\mathcal{D}_{XY}}(c) \leq 1/4\}.$$

By definition, with probability at least $1/2$ over the random draw of an i.i.d. sample of size $m_0$ from $\mathcal{D}_{XY}$ and the internal randomness of $\mathcal{A}$, the output of the algorithm belongs to $\mathscr{C}(\mathcal{D}_{XY})$. Thus, there exists a sample $S_0 \in (\mathcal{X} \times \{0,1\})^{m_0}$ such that

$$\Pr_{r\sim\mathcal{R}}[\mathcal{A}(S_0; r) \in \mathscr{C}(\mathcal{D}_{XY})] \geq 1/2.$$

---

[13]As in Gonen et al. [2019], the size is constant w.r.t. the accuracy, confidence parameters, but it might not be constant w.r.t. some parameter the describes the complexity of the concept class.

Although it is unclear how to identify such a $S_0$, notice that any sample $S \in (\mathcal{X} \times \{0,1\})^{m_0}$ is an $m_0$-neighbor of $S_0$. Since $\mathcal{A}$ is pure DP, it holds that

$$\Pr_{r \sim \mathcal{R}}[\mathcal{A}(S; r) \in \mathscr{C}(\mathcal{D}_{XY})] \geq 1/2 \cdot e^{-0.1 \cdot m_0} .$$

In particular, this holds for the dummy dataset $\bar{S}$ we fix before observing any data. Then if we sample $N = 2 \cdot e^{0.1 m_0} \cdot \log(3/\beta') = \Theta(\exp(m_0) \cdot \log(1/\beta'))$ i.i.d. random strings $r_1, \ldots, r_N$ from $\mathcal{R}$ i.i.d. the probability that none of the hypotheses $\{\mathcal{A}(\bar{S}; r_i)\}_{i \in [N]}$ is in $\mathscr{C}(\mathcal{D}_{XY})$ is at most

$$(1 - 1/2 \exp(-0.1 m_0))^{2 \exp(0.1 m_0) \cdot \log(1/\beta')} \leq \frac{\beta'}{3} .$$

We denote the event that one of the outputs is in $\mathscr{C}(\mathcal{D}_{XY})$ by $\mathcal{E}$ and we condition on it for the rest of the correctness proof. The next step is to replicably learn the finite concept class $\bar{\mathscr{C}} = \{\mathcal{A}(\bar{S}; r_1), \ldots, \mathcal{A}(\bar{S}; r_N)\}$ we have constructed using Theorem F.1 with accuracy $1/8$, confidence $\beta'/3$, and replicability $\rho$. Thus, we can see that $O\left(\frac{\log^2 N + \log \frac{1}{\beta \rho}}{\rho^2} \log^3 \frac{1}{\rho}\right) = \widetilde{O}\left(\frac{m_0^2 \log(1/\beta')}{\rho^2}\right)$ samples suffice for this task. Let $\mathcal{E}'$ be the event that this estimation is correct, which happens with probability at least $\beta'/3$. Conditioned on that event, outputting the hypothesis generated by the learner gives a solution with error at most $1/4 + 1/8 = 3/8$. Notice that the good events happen with probability at least $1 - 2\beta'/3 \geq 1 - \beta$, so the correctness condition is satisfied.

**Replicability.** Notice that the first step of the algorithm where we fix a dummy dataset $\bar{S}$ of size $m_0$ and run the algorithm on i.i.d. strings of its internal randomness is trivially replicable, since the dataset and the randomness are the same across the two executions. Then, by the guarantees of Theorem F.1, we know that the output of the algorithm will be the same across two executions with probability at least $1 - \rho$. Thus, overall we see that with probability at least $1 - \rho$, the output of our algorithm is the same across the two executions. $\qquad\square$

We emphasize that the number of oracle calls to the DP algorithm is constant with respect to the confidence and the correctness parameter, but could, potentially, be exponential with respect to some parameter that depends on the representation of the underlying concept class $\mathcal{X}$.

Equipped with the weak learner from Lemma F.2 we can boost its error parameter using the boosting algorithm that appears in Impagliazzo et al. [2022]. For completeness, we state the result below.

**Theorem F.3** (Replicable Boosting Algorithm [Impagliazzo et al., 2022])**.** *Let $\mathcal{D}_{XY}$ be the joint distribution over labeled examples as in Lemma F.2. Fix $\alpha, \rho, \beta' > 0$. Let $\mathcal{A}$ be a $\rho$-replicable $(\gamma, \beta)$-weak learner with sample complexity $m(\gamma, \rho, \beta)$, i.e., its error is at most $1/2 - \gamma$, with probability at least $1 - \beta$. Then, there exists an efficient $\rho$-replicable boosting algorithm $\mathcal{A}'$ such that with probability at least $1 - \beta'$ it outputs a hypothesis $h$ with $\mathrm{loss}_{\mathcal{D}_{XY}}(h) \leq \alpha$. The algorithm runs for $T = \widetilde{O}(\log(1/\beta)/(\alpha \rho^2))$ rounds and requires $T$ oracle calls to the weak learner. Moreover, it requires*

$$\widetilde{O}\left(\left(\frac{m(\gamma, \rho/6T, \beta)}{\alpha^2 \gamma^2} + \frac{1}{\rho^2 \alpha^3 \gamma^2}\right) \log \frac{1}{\beta'}\right)$$

*many samples, where $m(\gamma, \rho/6T, \beta)$ is the sample complexity needed for the weak learner to be $\rho/6T$-replicable and $(1/2 - \gamma, \beta)$-accurate.*

Combining the discussion above, we see that for any $\alpha, \rho, \beta \in (0, 1)^3$ we can transform a pure DP learner to a $\rho$-replicable $(\alpha, \beta)$-correct (strong) learner where the transformation is efficient with respect to the accuracy $\alpha$, confidence $\beta$, and replicability $\rho$. This proves Theorem 5.2, which we restate below for convenience.

**Theorem 5.2** (From Pure DP Learner to Replicable Learner)**.** *Let $\mathcal{X}$ be some input domain, $\mathcal{Y} = \{0, 1\}$, and $\mathcal{D}_{\mathcal{X}\mathcal{Y}}$ be a distribution on $\mathcal{X} \times \mathcal{Y}$ that is realizable with respect to some concept class $\mathscr{C}$. Let $\mathcal{A}$ be a pure DP learner that, for any $\alpha, \varepsilon, \beta \in (0, 1)$, needs $m(\alpha, \varepsilon, \beta, \mathscr{C}) = \mathrm{poly}(1/\alpha, 1/\varepsilon, \log(1/\beta), \dim(\mathscr{C}))$ i.i.d. samples from $\mathcal{D}_{\mathcal{X}\mathcal{Y}}$ and $\mathrm{poly}(m)$ running time to output a hypothesis that has error at most $\alpha$, with probability $1 - \beta$ in an $\varepsilon$-DP way. Then, for any $\alpha', \rho, \beta' \in (0, 1)$ there is a $\rho$-replicable learner $\mathcal{A}'$ that outputs a hypothesis with error at most $\alpha'$ with probability at least $1 - \beta'$ and requires $\mathrm{poly}(1/\alpha', 1/\rho, \log(1/\beta'), \dim(\mathscr{C}))$ i.i.d. samples from $\mathcal{D}_{XY}$ and $\mathrm{poly}(1/\alpha', 1/\rho, \log(1/\beta')) \cdot \exp(\dim(\mathscr{C}))$ running time.*

We reiterate that even though our transformation is efficient with respect to the input parameters $\alpha, \beta, \rho$, it might not be efficient with respect to some parameter that depends on the representation of the concept class. This is because the size of the "dummy" dataset $m_0$ could depend on the size of that representation and our transformation requires $\mathrm{poly}(m_0)$ samples but $\exp(m_0)$ running time. This is also the case for transformation from a DP learner to an online learner [Gonen et al., 2019].

