# OpenReview forum: "On the Computational Landscape of Replicable Learning"
_NeurIPS.cc/2024/Conference — NeurIPS 2024 poster_

### Official Review · Reviewer_9ean · 2024-06-29

**Soundness:** 1
**Presentation:** 3
**Contribution:** 3
**Rating:** 5
**Confidence:** 4

**Summary:**

The authors study the computational relationship between *replicable* learning algorithms, a recent notion of algorithmic stability [ILPS22] promising two runs of the algorithm over independent data output the same hypothesis with high probability assuming shared randomness, and other classical notions of algorithmic stability such as SQ-learning and differential privacy (as well as the closely related topic of online learning). The equivalences between such notions are well understood statistically, but this is the first paper to make a systematic attempt to study *computational* aspects of the problem.

More formally, the authors present both negative results (separations), and positive results (efficient transformations) between these notions:

On the negative side, the authors show:

1. A concept class based on one-way sequences which has no efficient online learner but can be learned replicably in polynomial time. This result is based on work of Bun separating private and online learning, but requires a new idea on the replicable side. In particular, two independent runs of the algorithm (with fresh samples) must identify a shared early element in the one-way sequence (using their shared randomness) to output the same unique hypothesis on the remainder. This is done via a new replicable quantile estimation lemma that leverages a variant of [ILPS22]'s randomized rounding methods.

2. A separation between SQ learning and replicable learning based on affine parities. While this class was already known to separate the two under the uniform distribution, no separation (or even efficient replicable algorithm) was known for more general distributions. Based on work of Blanc, Lange, Malik, and Tan, the authors give a general procedure to lift an efficient replicable algorithm for the uniform distribution for certain classes to a replicable algorithm for general distributions whose computational efficiency scales with the decision tree complexity. They then give examples of distributions with low decision tree complexity where the trivial `gaussian elimination’ based algorithm for the uniform case fails but their lifted algorithm runs in polynomial time (they also observe these distributions remain hard for SQ). This makes progress on a question of ILPS'22 about replicable algorithms where Gaussian elimination fails.

On the positive side:

1. There is an algorithm transforming any pure private learner for a class C to a $rho$-replicable learner in time exp(rep-dim(C))*poly(eps^{-1},\rho^{-1}), where eps is the accuracy parameter and rho is the replicability parameter. Their algorithm uses a classic procedure of Beimel Nissim and Stemmer which generates a finite approximation of C using a pure private learning, then learn this representation with the finite learner of [BGH+23]. To get polynomial dependence, this is run for constant error then boosted afterwords using replicable boosting.

**Strengths:**

Replicability is a critical notion in machine learning and the sciences in general, and has recently been a fruitful notion more generally in the study of algorithmic stability, leading e.g. to advances in differential privacy. Prior works in the area largely focus on the statistical complexity of replicable learning. Understanding the *computational* cost of replicability, both in general and as compared to other notions of stability, is a critical open problem and clearly of interest to the NeurIPS community. This paper is the first to initiate a systematic study of this problem, and makes initial progress on understanding connections with pure privacy, SQ, and online learning, standard notions in the literature.

Related to the above, the authors make progress on an open problem of [ILPS’22] to design efficient replicable algorithms for parities beyond the uniform distribution (namely in this case for distributions with constant decision tree complexity). This takes some work to formalize and is a reasonable contribution on its own from a computational standpoint.

**Weaknesses:**

The work has two main weaknesses.

First, the authors seem to have misunderstood prior separation results in the literature, and as a result, the presentation of `computational separations’ with respect to privacy in the paper (namely in Figure 1.1 and the exposition) is wrong. Namely, the authors claim that there is an “efficient transform from pure DP learning to replicable learning” and “no efficient transform from apx DP learning to replicable learning”, where “efficient” is in terms of eps and rho (error and replicability) but not the underlying dimension of the problem, but this seems false.

In particular, there actually *is* a transformation from approximate DP to replicability that is “efficient” in this sense. Correlated sampling can be run in time roughly scaling with the output domain of the private algorithm, then boosted via ILPS from constant accuracy/replicability in polynomial time in these parameters. In fact, *any* learning problem that is solvable replicably can be solved in time polynomial in eps and rho by boosting/amplifying, so “efficiency” in these parameters is not very meaningful. The question of efficiency instead should be one of domain-size/dimension, which (as the authors to their credit highlight several times) is not efficient in the given reduction.

Part of the confusion here seems to stem from the result of [BGH+23] giving a computational separation between apx DP and replicability. In Prop 5.1, the authors state [BGH+23] exhibit a PAC problem which is efficiently learnable under apx-DP, but cannot be replicable learned assuming one-way functions exist. As far as I can tell, this is not shown in [BGH+23]. First, the separation given by [BGH+23] is not for PAC-learning, it is for a somewhat contrived statistical task; separating the two in the PAC setting is open. Second, the separation has nothing to do with one way functions (which seems to be a different result in their paper), and relies on public key encryption. Third, the separation is in terms of the dimension/size of the space, not accuracy or replicability parameters.

The second weakness of this paper is that, while extra technical work is certainly required for several of the results in this paper (namely the online bound, and generalizing BLMT23), the ideas in this paper do not go substantially beyond known methods in the study of replicability. The online vs replicability result is not too much of a jump from its use in work of Bun separating privacy and online learning, (the new replicable quantile estimation method takes work but is fairly straightforward from techniques in ILPS22). The SQ/distribution-lifting result also follows largely from combining techniques of [ILPS22] with [BLMT23]’s non-replicable method which already relies mostly on statistical estimation sub-routines.

Overall, the authors have identified an important problem and made some nice partial progress in this front (including progress on an open problem of ILPS22), but combined with the issues above and without introducing substantially new ideas to the study of replicability I cannot recommend the work for acceptance in its current form.

**Questions:**

Given the above discussion of the apx-DP, perhaps a better way to justify the pure DP transformation would be to exhibit a sequence of concept classes over {0,1}^n with constant representation dimension for which the transformation from apx-DP is inefficient (or at least some evidence can be given for this)?

I will be happy to increase my scores if I have misunderstood something regarding these transformations in the paper and the authors can clarify, or if the authors agree and fix the current presentation.

POST REBUTTAL: The authors' proposed changes address my concerns in the soundness of the work, and I have updated my overall score accordingly.

---

> ### Author Rebuttal · Authors · 2024-08-03
>
> We thank reviewer 9ean for their thorough review, their suggestions regarding the presentation of the manuscript, and the clarification of results from prior work.
>
> We apologize for the confusion; indeed
> there was a misunderstanding on our end about the
> separation that [BGH+23] provided. We are committed to modifying
> the manuscript according to the reviewer's recommendation
> and have updated the figure in the "global" response section.
>
> Moreover,
> as the reviewer correctly points out, the transformation
> of [BGH+23] from approx DP to replicability requires
> exponential time in the size of the domain, and polynomial time
> in the approximation/confidence parameters. Thus, their
> result does not subsume ours. We also agree that the question
> the reviewer mentions is an interesting one, and we can
> mention it as an open problem.
>
> Regarding the second weakness,
> we believe
> the replicable lifting framework provides a non-trivial conceptual step towards understanding efficient replicable learnability beyond SQ algorithms,
> which is a direction we hope the community will further explore.
> Indeed, we provide a concrete
> distribution where
> we can obtain a replicable algorithm for learning parities using our framework,
> but Gaussian elimination, which is the standard
> algorithm in the absence of replicability, fails
> to be replicable. We view that as evidence
> that our transformation can indeed lead to replicable
> algorithms in settings where such results are lacking.
>
> We will include these discussions in the next iteration of our paper, and we apologize again for the misunderstanding.

---

> ### Comment · Reviewer_9ean · 2024-08-08
> **Rebuttal Followup**
>
> We thank the authors for their response and would like to ask a few follow-up questions before modifying the final review accordingly:
>
> First, it is unclear to me what the authors mean by saying the transformation in [BGH+23] is "exponential in the size of the domain". The correlated sampling transform should (at worst) scale inverse polynomially with the minimum probability of any output of the apx-DP algorithm. Often the size of this output space scales with the *dimension* of the problem, or can be made to by discretizing (see e.g. the application of [BGH+23]'s transform to Gaussian mean estimation in the same work). In fact, I'm not totally sure what the authors mean by domain here. Do you mean the output space of the Apx-DP algorithm? Or the original domain of the problem? I suppose in many cases  (e.g. in tabular settings) the size of the domain is exponential in dimension, so maybe this is what is meant. To be clear, I do not mean to claim here the Apx-DP reduction subsumes the one presented in this work. I agree this is almost certainly not true.
>
> Second, while the updated figure is certainly closer to accurate, I still find the diagram's representation of Apx-DP -> replicability vs Pure-DP -> replicability and its discussion of "computational separations" to be (unintentionally) misleading.
>
> For instance, the figure states that the dashed double arrow indicates "an efficient learner for a task... can be black-box transformed into an efficient learner for the same task... for a subset of the relevant parameters." In what sense does the correlated sampling transform fail this? As discussed above it can be made efficient in $\varepsilon,\delta$ and $\rho$, just like the Pure DP case, but may be expensive in dimension/output-space. Why does Apx-DP -> Replicability not fit the double dashed arrow as written?
>
> Conversely, the Pure DP case *could* still have a computational separation based on dimension. I.e. there could be a PAC problem that is privately learnable in polynomial time (in the ambient dimension), but requires exponential time replicably (under say cryptographic assumptions). This is exactly the sort of separation given in [BGH+23] (albeit not for a PAC problem), which as far as I can tell is cited as really the only reference for what a "computational separation" actually means in this figure. Why then is an efficient transform from Pure DP -> Replicability not listed as "open" if such a separation could still exist?
>
> Overall, I do not think the diagram's categories appropriately capture the known landscape of computational transforms.
>
> Could the authors please clarify if I am misunderstanding the diagram/results, or if not propose a change that reflects the above?

---

> > ### Author Response · Authors · 2024-08-09
> > **Response to Reviewer's 9ean Comments**
> >
> > Thank you again for your thorough comments. Let us
> > try to clarify the questions you ask.
> >
> > - Regarding the Apx-DP reduction to replicability:
> > Our understanding is that in order for [BGH+23] to use
> > correlated sampling it has to be the case that the output space
> > of the algorithm is finite.
> > To be more precise, based on [BGH+23], there is an efficient transformation from Apx-DP to perfectly generalizing algorithms. Next, the authors use correlated sampling to get a replicable learner.
> > Given a perfectly generalizing algorithm $A$ and sample $S$,
> > the correlated sampling strategy is applied to the distribution of outputs of
> > $A(S)$.
> > Hence, the output space of $A$ should be finite.
> > In the PAC learning setting,
> > in order to ensure that the algorithm
> > has finite output space,
> > one sufficient
> > condition is that the domain $\mathcal{X}$
> > is finite, and this is what our comment
> > was trying to state. To be even more
> > precise, for the case of PAC learning,
> > there is another transformation from apx-DP
> > to replicability that holds for
> > countable domains $\mathcal{X}$ that was
> > proposed by
> > [KKMV'23], but this approach i) goes
> > throgh the Littlestone dimension of the
> > class and ii) might not even be computable,
> > in its general form.
> > The correlated sampling step can
> > be explicitly implemented via rejection sampling from the output space of $A$.
> > The acceptance probability is controlled
> > by the pmf of $A(S)$, as the Reviewer mentions. As a result, in general, it is not computationally efficient.
> > For instance, if the finite input space to the correlated sampling strategy is $\\{0,1\\}^d$, then the runtime of the algorithm could be $\exp(d)$, since the acceptance probability is exponentially small in the dimension in the worst case. We will further clarify our comment to capture this behavior.
> >
> > - Figure: We will adjust the figure for the Apx-DP to Replicability as the Reviewer suggests and explicitly discuss the situation for the two directions.
> > For the Apx-DP to Replicability, we will add the above discussion and change the arrow type since the transformation is efficient in the other parameters except for $d$. For the PureDP to Replicability, we will clarify that it could still have a dimension-dependent computational separation, as the Reviewer comments.
> > Our intention was to depict the following
> > difference: both transformations from
> > apx-DP, pure-DP to replicability can
> > be efficient in $\varepsilon, \delta$ but,
> > in its general form, apx-DP to replicability
> > from [BGH+23] would need $\mathcal{X}$
> > to be finite (or some other condition that ensures the algorithm has finite output space), and apx-DP to replicability from [KKMV'23] for countable domains $\mathcal{X}$
> > might not even be computable, whereas
> > the pure-DP to replicability transformation
> > we provide has running time exponential
> > in the representation dimension.
> >
> > Please let us know if this answers your
> > questions or if we have missed some
> > point you were trying to make.

---

> > > ### Comment · Reviewer_9ean · 2024-08-11
> > > **Response**
> > >
> > > We thank the authors for the additional clarification and the explained differences make sense.
> > >
> > > It is definitely tricky to depict this in a simple diagram. I think the authors suggestion of expressing both as "efficient in some parameters" and explaining in more depth the differences is a reasonable option. Giving both different arrows w/r/t the infinite case could also work, though it might overcomplicate the diagram.
> > >
> > > I will update my review to reflect the authors' proposed changes.

---

### Official Review · Reviewer_iQWb · 2024-07-03

**Soundness:** 4
**Presentation:** 4
**Contribution:** 3
**Rating:** 6
**Confidence:** 4

**Summary:**

In the paper, the authors discuss the connection and differences among replicability, a novel stability condition for learning algorithms proposed by Impagliazzo et al. [2022], online learning, and differential privacy from a computational perspective. Their first contribution is a computational separation between online learning and replicable PAC learning. In particular, under standard cryptographic assumptions, there is a concept class that can be replicably learned, but no efficient online learning algorithm exists. The second contribution is a method to extend a replicable PAC learner that works under the uniform distribution to ones that work under more complex distributions, whose probability mass functions can be computed by decision trees. The final result is a way to transform a purely differentially private learner into a replicable one. Combining these with some existence hardness/equivalence results provides a figure (Figure 1.1) referred to by the authors as the "computational landscape of stability."

**Strengths:**

Figure 1.1 is an excellent summary of all known relationships between different notions of algorithmic stability. I believe the communities behind all three areas (replicable, online, DP) could benefit from such a roadmap. All the results are clean and well-motivated.
The one I like the most is the lifting result for replicable learner. Intuitively, it feels like replicable learners and statistical query algorithms are almost equivalent as most replicable algorithms we know are based on the idea of making each statistical query fired by the algorithm replicable. The only exception known before is learning parities under the uniform distribution over boolean hypercube. The authors demonstrate the possibility that there may potentially be much larger gaps between SQ algorithms and replicable learners.

**Weaknesses:**

While the results are conceptually novel and interesting, the techniques used are more or less standard. For example, the lifting result follows from building a replicable version of the routine from Blanc et al. [2023] for learning the decision tree structure of the distributions, and the core of it is just to estimate the influence of distributions in a replicable manner (via random statistical query rounding).

**Questions:**

As a pure DP algorithm is also an approximate DP algorithm by definition, I imagine that the reduction from Bun et al. [2023] could also transform a pure DP learner into a replicable learner. Could you elaborate more on how the performance of your reduction would differ from theirs quantitatively?

**Limitations:**

Yes.

---

> ### Author Rebuttal · Authors · 2024-08-03
>
> We thank reviewer iQWb for recognizing the contributions of our paper.
>
> The main result of [ILPS22] is that SQ-based algorithms can be made replicable. However, our understanding of replicability beyond SQ algorithms is fairly limited. The main candidate in understanding this question is learning parities. We make progress on this important problem by using the replicable lifting framework.
> This framework provides a non-trivial conceptual step towards understanding efficient replicable learnability beyond SQ algorithms,
> which is a direction we hope the community will further explore.
> Indeed, we provide a concrete
> distribution where
> we can obtain a replicable algorithm for learning parities using our framework,
> but Gaussian elimination, which is the standard
> algorithm in the absence of replicability, fails
> to be replicable. We view that as evidence
> that our transformation can indeed lead to replicable
> algorithms in settings where such results are lacking.
>
> It is, indeed, a correct
> observation that pure DP algorithms
> are also approximate DP algorithms.
> Thus, in
> principle, one could use
> the reduction from Bun et al. [2023].
> The catch is that this reduction
> is based on correlated sampling so it
> requires i) the output space of the algorithm to be finite and ii)
> even under finite output spaces, it needs
> exponential time in the size of that space.
>
> We will elaborate on these discussions in the next version of our work.

---

### Official Review · Reviewer_npit · 2024-07-13

**Soundness:** 3
**Presentation:** 4
**Contribution:** 3
**Rating:** 7
**Confidence:** 3

**Summary:**

Replicability is a notion of stability for learning algorithms, recently proposed by Impagliazzo et al. [ILPS22] to address the replicability crisis pervasive in scientific studies using statistical procedures. A learning algorithm $A$ is a function that takes as inputs a dataset $S \in (\mathcal{X} \times \mathcal{Y})^*$ and a (random) string $r \in \\{0,1\\}^*$, and outputs a hypothesis $f: \mathcal{X} \to \mathcal{Y}$. The algorithm is replicable if for independent draws of same sized datasets $S, S’$ and random $r$, $A(S, r) = A(S’, r)$ with high probability. Put simply, if two scientific labs use the same replicable algorithm to analyze independent datasets S and S’, and arrive at different conclusions, they will have a hard time blaming statistical fluctuations in the data.

This paper presents a variety of results that connect replicability to other well-studied concepts in learning theory, such as (realizable) online learnability and private learning. In particular, the authors show a computational separation between replicability and **online learnability**, assuming the existence of one-way functions. They also present a lifting procedure that transforms an efficient replicable PAC learner for a hypothesis class $\mathcal{H}$ over the uniform distribution on $\\{\pm 1\\}^n$ to replicable learners over any marginal distribution whose sample and time complexity depends on the complexity of the marginal. This lifting procedure is then used to design an efficient replicable algorithms for **learning parities over distributions that are far from uniform**. Furthermore, they show that any **pure DP learner** can be turned into a replicable one in time polynomial in all relevant problem parameters, except for the "representation dimension" of the hypothesis class.

**Strengths:**

This is a very well-written paper with solid technical contributions. Though the paper touches on various concepts in learning theory, the polished presentation ensures that the reader is not overwhelmed by the breadth of coverage. As replicability is a relatively new concept in learning theory, I was not familiar with it. However, the connections between replicability and other concepts are rather surprising and pleasing to someone new to the topic. The clear and well-organized proofs, each employing diverse techniques, were relatively easy to follow.

I found the application of the lifting procedure to replicable parity learning and the distinction from Gaussian elimination particularly interesting. For the uniform distribution, it is straightforward to see that Gaussian elimination is replicable for data labeled by parity functions. However, one can define simple distributions on which Gaussian elimination is *not* replicable (though the zero-one loss *is* small with high probability). This highlights the necessity and the non-triviality of the lifting procedure for efficient replicable learners over the uniform marginal. It also shows that computational separation between replicable learning and statistical query (SQ) learning extends to non-uniform marginal distributions.

**Weaknesses:**

Given that this is a solid and well-polished paper, I did not find any significant weaknesses, only a few minor points below.

- **Elaborating replicability with simplified, realistic statistical examples.** The paper begins by addressing the replicability crisis in scientific fields, but once the authors define replicability within the learning theory setup, the initial narrative gets lost. What would the random strings model in a real-world scientific study? Does it model the PRG seed number that experimenters use in their PyTorch code? Taking a simplified but realistic example (e.g., a hypothetical FDA approval procedure based on clinical data collected by independent labs), and mapping the A's, S's and r's to real-world concepts would be helpful.

- **Ambiguity in the big questions.** I found some of the "big" questions Q1-Q4 [page 2-3] too generic to be useful. In particular, questions like "How does replicable learning *relate to* online/private learning?" are extremely underspecified because "relate to" can have multiple interpretations. It would have been more helpful if the authors posed more specific motivating questions, such as "Is there a computational separation between replicably learnable classes and online learnable classes?"

**Questions:**

- What is the "representation dimension" of a hypothesis in Theorem 5.2?

- Can we define replicability with continuous random sources? Or are there obstacles to meaningfully defining replicability with respect to continuous sources?

---

> ### Author Rebuttal · Authors · 2024-08-03
>
> We thank reviewer npit for the insightful suggestions regarding the presentation of our paper.
>
> The random string in the definition of replicability does indeed model the random seed of a learning algorithm in practice
> and sharing internal randomness can be easily implemented in practice by sharing the random seed.
> We can indeed define
> replicability with continuous
> random sources, but to make
> the presentation easier to follow and tie
> it to real-world settings,
> we decided to stick with random binary strings.
> We will provide other concrete examples for the definition in the next iteration of our manuscript.
>
> We will also restate the motivating questions with more specific language.
>
> The representation dimension is a combinatorial dimension, similar to VC dimension, that characterizes which classes are PAC learnable by pure DP algorithms. We will formally give its definition in the next revision of our work and mention how it has been used
> in prior works on pure DP.

---

> > ### Comment · Reviewer_npit · 2024-08-09
> >
> > Thank you for addressing my questions and being receptive. My rating of the paper remains the same.

---

### Official Review · Reviewer_eJSY · 2024-07-14

**Soundness:** 4
**Presentation:** 4
**Contribution:** 3
**Rating:** 6
**Confidence:** 4

**Summary:**

This work contributes to the recently evolving area of replicable learning. This work establishes three main results

1. It is known that online learning algorithms can be replicable and replicable learning algorithms yield online learning algorithms. The work focuses on the computational complexity of these transformations and establishes a negative result--there exist concept classes that are efficiently, replicably, and learnable but there are no efficient online learning algorithms (assuming the existence of one-way functions).
2. Recent work showed that PAC learning algorithms under the uniform distribution can be black-box converted into distribution-free PAC learning algorithms. This work shows the transformation can be made replicable.
3. It is known there exist concept classes that are approximately DP learnable, but not efficiently, replicable learnable.  This work shows that if the concept class is pure DP-learnable, then it is efficiently replicable learnable.

**Strengths:**

This solid work clarifies several questions on replicable learning in the context of computational efficiency. The paper is very well written and the proofs are rigorous.

**Weaknesses:**

Main weakness is that obtained by combining known works. It is not clear to be that there is not that much novelty in the proofs.  for example,

1. Proof of Theorem  2.1 proceeds in two parts.  a) existence of a concept class that is not online learnable, b) designing a replicable learning algorithm for this class. (a) is known by prior works, (b) replicable algorithm follows by the (now) standard technique of random thresholding/rounding.
2. Please see Q2.

Though the proofs may not be novel, the final results that the authors obtain are interesting.

**Questions:**

Pertains to the weakness. Can you explain the new ideas needed to obtain the results?

1. ILPS22 presents a replicable algorithm for an approximate median. The approximate quantile algorithm seems to build on similar ideas. What are the new ideas in the proof?
2. Similarly, the critical ingredient in the proof of 3.2 is the replicable influence estimation. Once we have a replicable influence estimation the proof of Blanc et al goes through with minor changes. The design of replicable influence estimatior follows by the randomized rounding technique of ILPS22.  Are there any technical/conceptual challenges?

---

> ### Author Rebuttal · Authors · 2024-08-03
>
> We thank reviewer eJSY for recognizing the importance of our results
> and their suggestions regarding the presentation of our paper.
>
> We believe that the main contribution of our work, as the reviewer acknowledges, is a collection of interesting conceptual results that are missing from prior work. For instance, while the replicable influence estimator is not overly technically involved, it constitutes a key ingredient for the use of the lifting framework, which we find conceptually interesting. To justify this further,
> the main result of ILPS22 is that SQ-based algorithms can be made replicable. However, our understanding of replicability beyond SQ algorithms is fairly limited. The main candidate in understanding this question is learning parities.
> It is clear that under the uniform marginal distribution (or similar nice distributions) the problem is trivially replicable; yet beyond such distributions, it was not clear how to argue about replicable algorithm design for parities.
> We make progress on this important problem by using the replicable lifting framework.
> This framework provides a non-trivial conceptual step towards understanding efficient replicable learnability beyond SQ algorithms,
> which is a direction we hope the community will further explore.
> Indeed, we provide a concrete
> distribution where
> we can obtain a replicable algorithm for learning parities using our framework,
> but Gaussian elimination, which is the standard
> algorithm in the absence of replicability, fails
> to be replicable. We view that as evidence
> that our transformation can indeed lead to replicable
> algorithms in settings where such results are lacking.
>
> At a more technical note, our replicable quantile estimation differs from the replicable median algorithm of [ILPS22] in at least two ways.
> Firstly,
> the replicable median algorithm of [ILPS22] seems to rely heavily on properties of (approximate) medians in order to satisfy the approximation guarantees.
> On the other hand,
> our algorithm works regardless of the desired quantile.
> Secondly,
> their median algorithm relies on a non-trivial recursive procedure
> while our replicable quantile algorithm is considerably simpler and is based on a concentration of the CDF through the DKW inequality.
>
> We will include these discussions in the next version of our manuscript.

---

### Author Rebuttal · Authors · 2024-08-03

Following reviewer 9ean's suggestions,
we have uploaded a slightly modified figure
of the computational landscape of stability to clarify the computational separation between approximate DP and replicability given in [BGH+23],
which is not for PAC learning but for some other statistical task.
We will also modify the exposition accordingly.

---

### Decision · Program_Chairs · 2024-09-25

**Decision:**

Accept (poster)

**Comment:**

This work presents several results on the computational complexity of replicable PAC learning, a recently introduced notion of stability of learning. In particular, it gives two separation of efficient replicable learning from online learning and SQ learning. It also discusses the relationship of the notion to differentially private learning although, as pointed out by reviewers, the initial claims were based on a number of misunderstandings. Overall this is nice technical progress even if a purely theoretical setting is likely to substantially limit the appeal of the work.